# Conspecific and heterospecific grass litter effects on seedling emergence and growth in ragwort (*Jacobaea vulgaris*)

**Henrike Möhler**⦿*, **Tim Diekötter, Geeltje Marie Bauer, Tobias W. Donath**

Department of Landscape Ecology, Institute for Natural Resource Conservation, Kiel University, Kiel, Germany

* hmoehler@ecology.uni-kiel.de

## Abstract

*Jacobaea vulgaris* Gaertn. or common ragwort is a widespread noxious grassland weed that is subject to different regulation measures worldwide. Seedling emergence and growth are the most crucial stages for most plants during their life cycle. Therefore, heterospecific grass or conspecific ragwort litter as well as soil-mediated effects may be of relevance for ragwort control. Our study examines the effects of conspecific and heterospecific litter as well as ragwort conditioned soil on seedling emergence and growth. We conducted pot experiments to estimate the influence of soil conditioning (with, without ragwort), litter type (grass, ragwort, grass-ragwort-mix) and amount (200 g/m², 400 g/m²) on *J. vulgaris* recruitment. As response parameters, we assessed seedling number, biomass, height and number of seedling leaves. We found that 200 g/m² grass litter led to higher seedling numbers, while litter composed of *J. vulgaris* reduced seedling emergence. Litter amounts of 400 g/m² had negative effects on the number of seedlings regardless of the litter type. Results for biomass, plant height and leaf number showed opposing patterns to seedling numbers. Seedlings in pots treated with high litter amounts and seedlings in ragwort litter became heavier, grew higher and had more leaves. Significant effects of the soil conditioned by ragwort on seedling emergence and growth were negligible. The study confirms that the amount and composition of litter strongly affect seedling emergence and growth of *J. vulgaris*. Moreover, while conspecific litter and high litter amounts negatively affected early seedling development in ragwort, those seedlings that survived accumulated more biomass and got taller than seedlings grown in heterospecific or less dense litter. Therefore, ragwort litter has negative effects in ragwort germination, but positive effects in ragwort growth. Thus, leaving ragwort litter on pastures will not reduce ragwort establishment and growth and cannot be used as management tool.

## Introduction

*Jacobaea vulgaris* Gaertn. or common ragwort is a widespread noxious grassland weed that is native to Eurasia, and invasive in North America, New Zealand, and Australia. Since the

**Data Availability Statement:** All relevant data are within the manuscript and its Supporting Information files.

**Funding:** This work was supported by the Stiftung Naturschutz Schleswig-Holstein (http://www. stiftungsland.de). The funders had no role in study design, data collection and analysis, decision to publish, or preparation of the manuscript.

**Competing interests:** The authors have declared that no competing interests exist.

pyrrolizidine alkaloids of the plant pose a health risk to cattle that consume this plant [1], its occurrence in high densities is a challenge for grassland management. Consequently, control of common ragwort is prescribed in the UK, Ireland, New Zealand, Australia, and Friesland, province of the Netherlands [2]. Therefore, several studies have aimed to find effective and efficient ways to manage this grassland weed [2–5]. Some of these studies came to the conclusion that specific cutting regimes are a promising approach [6]. Biomass containing toxic ragwort cannot be used as hay or silage and the high number of long-living seeds rules out composting biomass [7, 8]. Consequently, the biomass is often disposed of as special waste [9]. However, cut ragwort litter is often left on the grassland, because removing the litter is laborious and costly. Yet, the occurrence of conspecific allelopathic effects from litter leachates [10, 11] suggests that simply leaving ragwort litter on mown grassland could be an additional component in managing ragwort abundance. As a first step to evaluate the validity of this approach, we experimentally tested this management option.

Litter influences micro-environmental conditions, effects nutrient cycling and can have effects on seedling germination and growth via mechanical and chemical effects [12, 13]. Positive litter effects are due to higher soil humidity and a lower amplitude of diurnal temperature fluctuations beneath a low litter cover [14]. Still the size of litter effects on seedling germination and establishment do also depend on the amount of litter present. If litter layer gets thicker, it reduces the amount of light that reaches the seeds. However, seeds of ragwort depend on a certain light amount to germinate [15]. On the other side germination of ragwort was facilitated under light sand cover compared to no cover [15]. In addition, thick litter layers that preserve a relative high humidity can facilitate the spread of mould that may cause decay of seeds [16]. After germination seedlings can be hampered in their growth or even die if litter amounts are too high. Thus Van der Meijden and van der Waals-Kol found significantly fewer seedlings in pots with high vegetation cover [15]. Litter may also exert chemical effects by nutrient immobilization, depletion of oxygen in the soil or toxicity of carbon dioxide produced by decomposers or by releasing phytotoxins [11, 14, 17]. Through these mechanisms litter potentially influences community organization via delaying and reducing or facilitating seedling emergence or seedling growth [13, 18].

Litter type present is another important factor that can influence whether seed germination and seedling establishment is facilitated or hampered [19]. Litter type changes conditions for germination and growth as chemical as well as physical structure depend on the characteristics of species [19]. Grass litter, for example, might be less hampering than large-leafed forb litter due to a lower amount of phytotoxins and an easier to penetrate structure [17, 19]. On the other side, forb litter often decomposes faster thereby reducing physical constrains for seedling emergence. Under natural conditions, however, litter is not usually monospecific, e.g. in grasslands it consists of a mixture of forb and grass litter. The effects of litter from plant communities are driven by species-specific litter properties [20] and may either be purely additive as hypothesized in the mass-ratio-hypothesis [21] or non-additive, as litter types may interact. In this context, Gartner and Cardon [22] found that more than half of the litter mixtures they reviewed behaved in a non-additive way concerning factors such as mass loss, nutrient transfer or decomposer composition.

Hypothetically conspecific litter may facilitate the establishment of conspecific seeds compared to heterospecific litter that may inhibit the growth of other species through allelopathic effects [23]. Yet, as intraspecific competition can be more intense than interspecific competition [24, 25], inhibiting effects of conspecifics may also be expected. While many studies have shown facilitating effects of adult plants on conspecific seedlings [26–28], particularly in weed species [29–31], there are also studies demonstrating the opposite [32, 33]. In line with the latter, studies on ragwort showed the negative autotoxic potential of ragwort litter and soil near

to ragwort plants [11, 34]. Also, Bezemer et al. 2006 and Van de Voorde et al. 2011 found a negative intraspecific plant-soil-feedback for ragwort, which they assumed is probably driven by pathogens that accumulate in the soil community [35, 36]. Under these circumstances, negative effects of ragwort litter and ragwort conditioned soil on ragwort performance (e.g. lower biomass) may either be additive or not. However, the influence of litter during seedling emergence might be stronger than during seedling growth, since at the latter stage impacts of competition are of higher importance than direct litter effects [37]. Thus, if litter reduces seedling density through self-thinning, surviving seedlings might have an advantage later on, as intraspecific competition would be reduced [38].

The present study is to our knowledge the first to test the influence of the combined effects of litter amount, type and soil conditioning on seedling emergence and growth in ragwort. Therefore, we set up a common garden experiment, where we assessed the conspecific and heterospecific effects of litter on the number of emerged ragwort seedlings, their biomass, height, leaf number and specific leaf area. We posed the following questions:

I.  Does litter type influence the emergence and growth of ragwort seedlings?
    We expect lower numbers of ragwort seedlings to emerge from beneath conspecific litter compared to pure grass litter or a mixture of grass and ragwort litter. As lower competition results in stronger growth we expect converse patterns to occur in seedling growth after emergence.

II. Does litter amount influence seedling emergence and growth of ragwort and does this depend on litter type?
    We expect negative litter effects to be stronger with increasing litter amount. We also expect the size of litter effects to increase with the portion of ragwort litter in the litter cover. Seedling performance is hypothesized to be higher under medium ragwort amounts and highest under low amounts of pure grass litter.

III. Does ragwort-conditioned soil influence conspecific seedling emergence and growth? If so, do these effects interact with the litter effects?
    We hypothesized an inhibitory effect of ragwort-conditioned soil on its own emergence and growth. We expect these effects to be additive for ragwort litter.

## Material and methods

### Study species

*Jacobaea vulgaris* L. syn. *Senecio jacobaea* Gaertn. or common ragwort is an herbaceous monocarpic plant from the Asteraceae family [39]. It is a hemicryptophyte, forming a rosette in the first year and flowering in the second year [39]. It prefers open patches and disturbed sites. Thus, it occurs on ruderal sites, along roadsides, pastures and grassland ecosystems in most temperate regions of the world. It is often regarded as a noxious weed due to its content of toxic pyrrolizidine alkaloids [2]. Ragwort shows ecological characteristics of a pioneer species, in particular its numerous light seeds and fast growth enable ragwort to spread rapidly in open areas [1]. Each plant produces several thousand seeds per year [1]. Seeds are mainly dispersed by wind over a distance of usually less than 15 m [40]. Seed survival (up to 20 years) and germination rates (about 80%) are high [41].

### Experimental design

We studied the effects of *litter amount* (200 (low) vs. 400 (high) g/m$^2$), *litter type* (ragwort litter, grass litter, 1:1 mix of ragwort and grass litter) and *conditioned soil* (soil conditioned with ragwort vs. unconditioned soil that is soil not conditioned by ragwort) on seedling emergence

and growth. The litter amounts applied corresponded to the range of average productivity of extensively managed grasslands in Northern Germany [42]. Loydi et al. [11] found positive litter effects to occur at these litter amounts. Thus, the litter amounts applied represent litter quantities that ragwort seeds are exposed to naturally and that seem to affect germination.

Seeds of *J. vulgaris* were hand-collected from at least 20 plant individuals on each of eight extensively managed pastures across Schleswig-Holstein from August to September 2015 and mixed for the experiment. These seeds were dry stored in darkness at room temperature until sowing on 21st April 2016. An initial viability test with a 1% tetrazolium chloride solution [43] on additional seed batches showed that most seeds (> 80%*)* were viable. *J. vulgaris* litter and conditioned and unconditioned soil was collected from rosettes on a ragwort-infested extensive cattle pasture near Preetz, Germany (54.227175, 10.242809) in April 2016. We assumed soil to be affected by ragwort, i.e. conditioned, if at least four ragwort-plants were growing within 0.5 m$^2$. We refer to unconditioned soil when soil was taken from the center of an area (diameter 10 m) where no ragwort was growing. Conspecific litter was collected, cleaned and dried for two days at room temperature and for 24 hours at 40°C to constant weight. This way we ensured that chemicals potentially affecting germination were not changed. For heterospecific grass litter, we used dried plant material from a grassland mainly consisting of sweet grass (Poaceae) species.

We set up a completely randomized experimental design. We prepared pots of 1 dm$^3$ volume (c. 10 × 10 × 10 cm) for each litter amount × litter type × soil conditioned (12 combinations) replicated eight times. To assess for seedlings emerging from the soil taken from ragwort stands we additionally included 20 extra-control pots, ten with ragwort conditioned soil and ten filled with unconditioned soil. In none of these extra control-pots ragwort seedlings emerged. Half of all the pots were filled with a 6:1 mixture of commercial potting soil (Fruhstorfer Erde®, Type P, Industrie-Erdenwerke Archut GmbH, Germany) and ragwort-conditioned soil or unconditioned soil, respectively. We followed the protocol of van de Voorde et al. [44] concerning inoculation and mixing of soil, as it led to significant differences between conditioned and unconditioned soil in their experiment.

In late April 2016, we sowed 50 seeds in each experimental pot and added litter on top of it. Pots were placed randomly in a common garden in Kiel, Germany (54.346883, 10.106829), surrounded by another row of pots (to reduce desiccation) and protected by a light metal grid (mesh size 7 cm x 7cm) to avoid litter being blown away. Pots were kept constantly moist by regular watering.

Germination started in mid May 2016. Seedlings that emerged above the litter cover were counted weekly until the tenth week after the experiment started. After the tenth week, we counted the total number of leaves per plant and measured the height of the tallest plant in the pot. We determined specific leaf area for five pots per combination by cutting the youngest fully developed leaves of three plants. These were scanned and leaf area in mm$^2$ measured by the programme IMAGEJ [45]. Afterwards leaves were dried at 60°C and weighed to the closest milligram. The same was done with total aboveground biomass.

## Data analysis

We tested for differences in number of emerged seedlings and measurements of seedling growth represented by: total biomass per pot and per plant, plant height, leaf number and specific leaf area. We used a three-factorial ANOVA to analyze the effect of litter type, litter amount and soil type, and their interactions. Residuals were examined visually to assess validity of assumptions for analysis of variance, i.e. normal distribution and variance homogeneity. Data for biomass per pot and biomass per plant had to be ln transformed to meet assumptions

of analysis of variance. Subsequent to significant ANOVA results, post-hoc Tukey-HSD test was employed to test for individual differences. In addition, we used contrasts to specifically test whether the effect of mixed litter on response variables was mid-way between that of grass and ragwort alone (additive effect of pure type litters in the mixture). Since specific effects of the different litter types on seedling emergence and growth of ragwort, rather than general litter effects, were the focus in our study, we did not include the controls in our model but included them for visual comparison in the figures. In addition, we analyzed the controls for soil effects as well as for differences to the litter types across litter amount using contrasts. As a measure for the relative contribution of each factor and their interactions to the total variability of the response parameters of interest, we used the ratio of the sum of squares of the factor or interaction of interest to the total sum of squares (i.e. for all factors, their interactions and error). All analyses were done using R version 3.6.1 [46].

## Results

In general, our results showed significant main effects of litter type and litter amount on all response variables: seedling number, biomass per plant and pot, plant height and mean leaf number. In contrast, the soil factor (conditioned vs. unconditioned) was not significant for any response variables under the litter treatments. Only the case of the separate analysis of controls (seeds added but no litter) soil revealed a significant effect on plant height. Our analyses revealed two significant interactions between main factors. In the case of biomass per plant and pot and seedling height we found a significant two-way *litter type × litter amount* interaction. Effects of the treatments on specific leaf area were not significant (results not shown).

### Seedling emergence

We found that less seedlings emerged in ragwort litter compared to grass litter. Number of emerged seedlings differed significantly between all three litter types (Fig 1). In addition, contrast analyses of the control against litter types revealed that the litter mixture had no significant effect on seedling number compared to the control (p = 0.30), while in the control seedling number differed significantly from grass and ragwort litter (p = 0.016 and p ≤ 0.0001, respectively). From beneath grass litter, about 50% more seedlings emerged compared to ragwort litter (18.1 ± 1.0 vs. 27.6 ± 0.8), while emergence from beneath mixed litter was intermediate (22.8 ± 0.8), roughly 25% greater than in pure ragwort (Fig 1). Litter effects were additive, i.e. the effect size of mixed grass-ragwort litter did not deviate significantly from the expected mean effect of pure grass and ragwort litter (p = 0.99).

High litter amounts decreased seedling emergence. Across litter types higher litter amounts of 400 g/m$^2$, significantly reduced seedling numbers by about one fifth, compared to low litter cover of 200 g/m$^2$ (24.2 ± 1.0 vs. 20± 1) (Fig 1).

While both, litter type and litter amount, affected seedling emergence significantly, the amount of explained variance was more than twice as high for litter type than litter amount (38.22% vs. 16.58%; Table 1).

### Seedling growth

Across litter amounts seedlings in ragwort litter had twice the biomass per pot as those grown in grass litter (1.1 ± 0.1 g vs. 0.4 ± 0.1 g) (Fig 2). Interestingly, seedling biomass per pot in the litter mixture (0.9 ± 0.1 g) lay between that of seedlings growing in pure grass or ragwort litter but in contrast to seedling emergence, the litter type effects were not additive, i.e. seedling biomass per pot in the mixed litter treatment lay significantly above the mean of seedling biomass in grass and ragwort litter (p = 0.001, Fig 2). Across litter types, biomass per plant and per pot

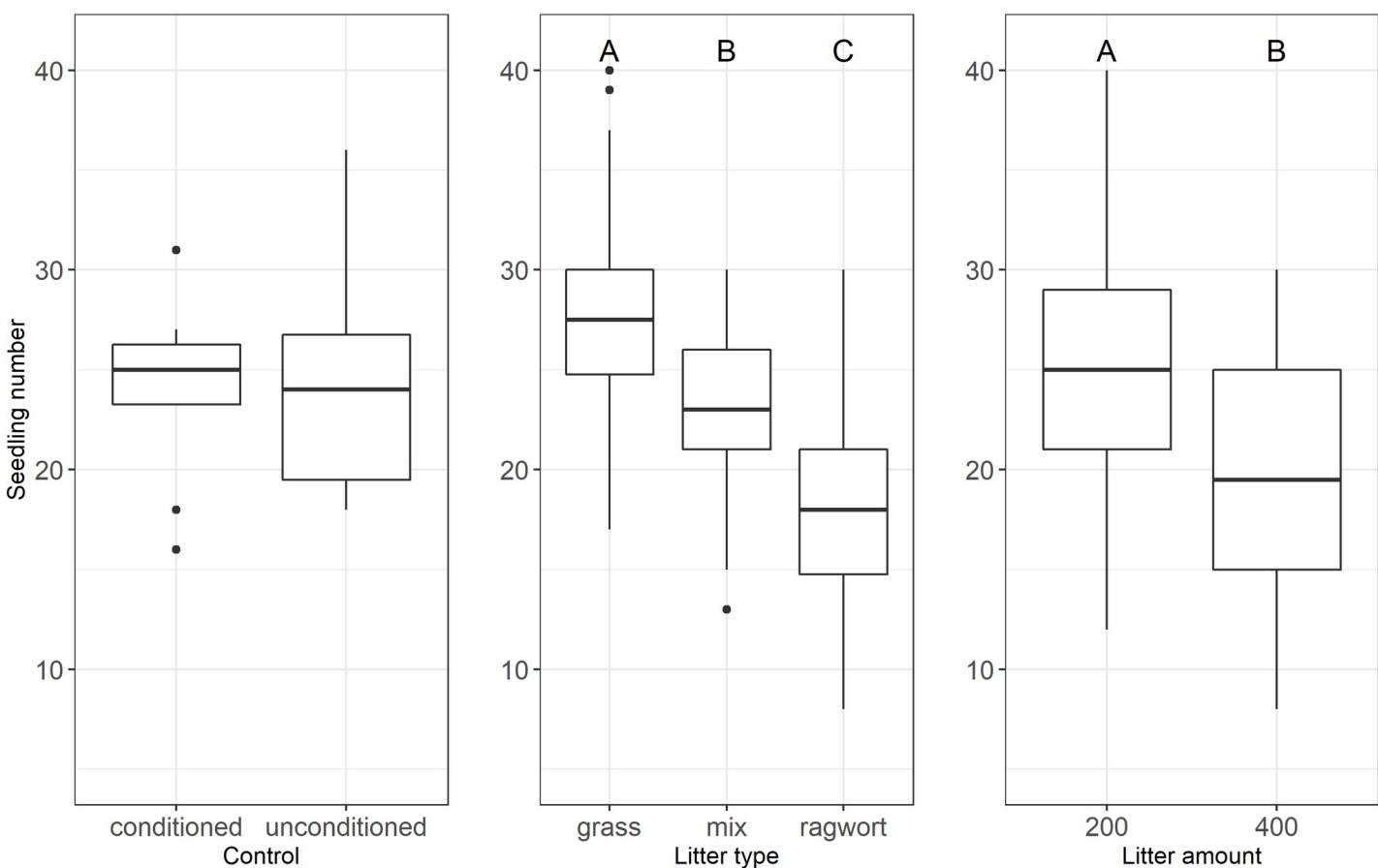

**Fig 1. Seedling number: Effects of soil conditioning on the control (left), litter type (grass, mix, ragwort (middle)) and different litter amounts (200 g/m² litter and 400 g/m² (right)).** In the box plots middle lines represent median, boxes represent the first and third quartiles, lower and upper bars represent the minimum and the maximum and points represent outliers (i.e. points above 1.5 SD). Upper case letters indicate significant main effects between treatments (TukeyHSD, P ≤ 0.05).

**Table 1. Results of a three-way ANOVA on the effects of litter type, litter amount and soil on seedling number and measures of seedling growth (biomass per plant, plant height and mean leaf number).**

| Source of variation | df | Seedling number | | | | Ln Biomass per pot | | | | Height | | | | Mean leaf number | | | |
|---|---|---|---|---|---|---|---|---|---|---|---|---|---|---|---|---|---|
| | | MSQ | F | P | vc | MSQ | F | P | vc | MSQ | F | P | vc | MSQ | F | P | vc |
| **Litter type** | 2 | **726.8** | **38.107** | **≤ 0.0001** | **38.22** | **10.824** | **95.940** | **≤ 0.0001** | **55.23** | **234.63** | **98.618** | **≤ 0.0001** | **56.51** | **1.602** | **7.868** | **≤ 0.0001** | **11.64** |
| **Litter amount** | 1 | **630.4** | **33.053** | **≤ 0.0001** | **16.58** | **12.355** | **5.415** | **≤ 0.0001** | **13.82** | **105.84** | **44.486** | **≤ 0.0001** | **12.74** | **5.505** | **27.037** | **≤ 0.0001** | **20.02** |
| **Soil** | 1 | 42.7 | 2.237 | 0.138 | 1.12 | 0.008 | 0.072 | 0.427 | 0.18 | 0.07 | 0.030 | 0.864 | 0.01 | 0.108 | 0.532 | 0.468 | 0.40 |
| **Litter type x litter amount** | 2 | 52.3 | 1.371 | 0.259 | 1.38 | **1.252** | **1.127** | **≤ 0.0001** | **5.75** | **21.84** | **9.181** | **≤ 0.0001** | **5.26** | 0.542 | 2.663 | 0.076 | 3.94 |
| **Litter type x soil** | 2 | 21.3 | 0.558 | 0.575 | 0.56 | 0.043 | 0.125 | 0.333 | 0.64 | 0.14 | 0.059 | 0.943 | 0.04 | 0.104 | 0.512 | 0.601 | 0.75 |
| **Litter amount x soil** | 1 | 1.0 | 0.055 | 0.816 | 0.03 | 0.004 | 0.001 | 0.924 | 0.00 | 3.15 | 1.326 | 0.253 | 0.40 | 0.012 | 0.059 | 0.809 | 0.04 |
| **Litter type x litter amount x soil** | 2 | 0.1 | 0.1 | 0.004 | 0.00 | 0.049 | 0.038 | 0.336 | 0.19 | 3.99 | 1.679 | 0.192 | 0.96 | 0.139 | 0.684 | 0.507 | 1.01 |
| **Error** | 84 | 1602.0 | | | 42.12 | 0.104 | | | 24.18 | 2.38 | | | 24.07 | 0.204 | | | 62.18 |

d.f. = degrees of freedom; MS = mean sum of squares; vc (%) = relative contribution of individual factors and their interactions to total variation. Significant effects *(P < 0.05)* are given in bold.

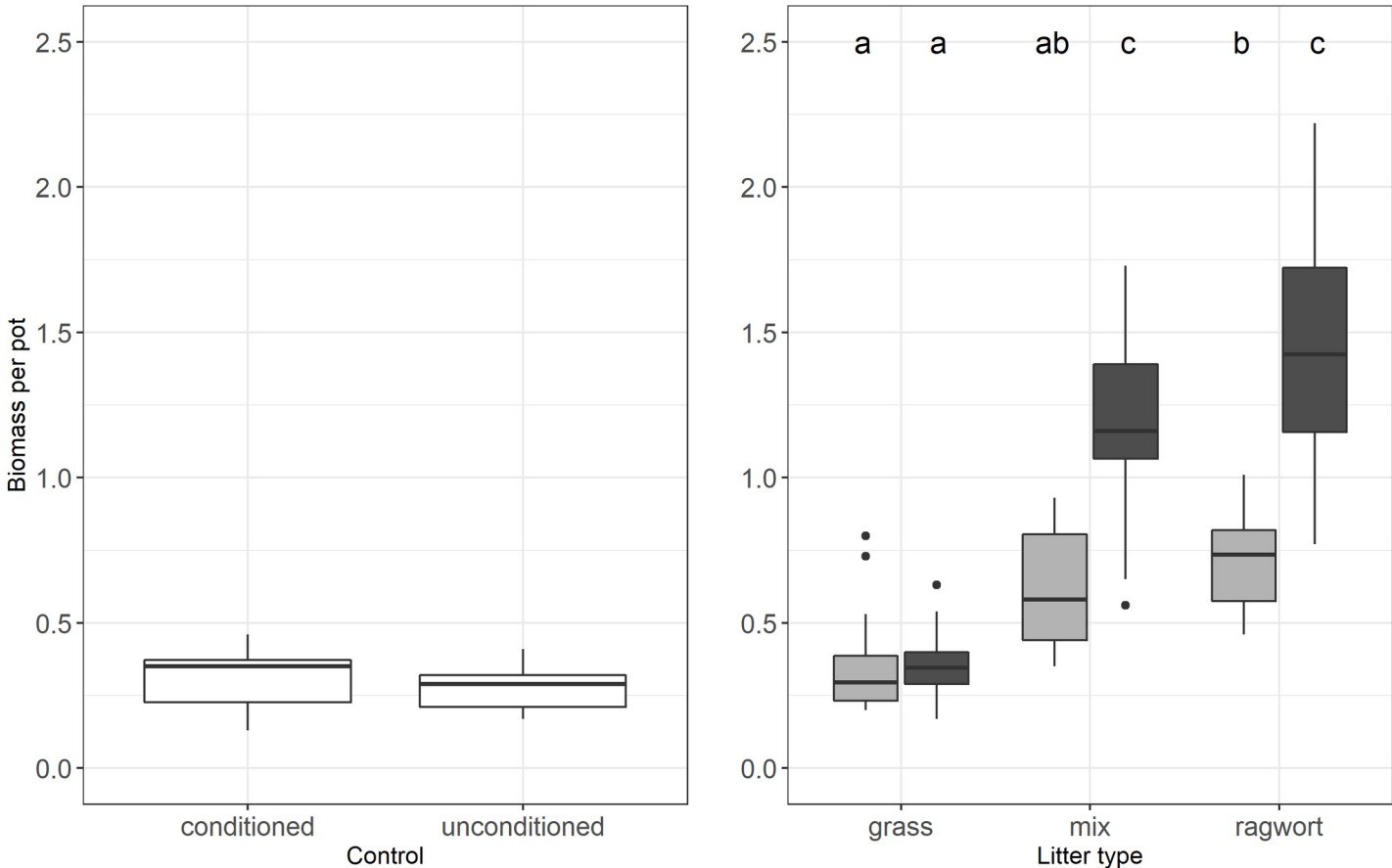

**Fig 2. Biomass per pot—Effects of soil conditioning on the control and the two-way interaction of litter type and litter amount.** In the box plots middle lines represent median, boxes represent the first and third quartiles, lower and upper bars represent the minimum and the maximum and points represent outliers (i.e. points above 1.5 SD). Grey: 200 g/m$^2$ litter, black: 400 g/m$^2$ litter. Lower case letters indicate significant interactions between treatments (TukeyHSD, P≤ 0.05).

increased with increasing litter amount applied (0.56 ± 0.03 g in low litter vs. 1.00 ± 0.08 g in high litter). Contrast analyses of the control against the three litter types revealed no significant difference of biomass per pot between the control and grass litter (p = 0.40) but significant differences in biomass between control and litter mixture as well as ragwort litter (both p ≤ 0.0001). Seedling biomass per plant showed almost the same response patterns but the litter effect was additive (p = 0.88; S1 Fig).

More than half of the variance in biomass per pot was explained by litter type, whereas litter amount only explained around one tenth of the variance (Table 1). The significant interaction, which accounted only for 6% of the variance, was caused by increased biomass with increasing litter amount when litter contained ragwort compared to no effect of amount in grass litter (Table 1 and Fig 2).

Seedlings grew tallest in high amounts of ragwort litter (Fig 3). Across amounts seedlings in ragwort litter grew almost three times as tall as in grass litter (8.1 ± 2.8 cm vs. 2.8 ± 0.9 cm) and about one third taller compared to the litter mixture (6.5 ± 0.3 cm). Litter type influenced plant height significantly, as did litter amount and the interaction of these factors (Fig 3). More than half of the variance was explained by litter type (Table 1). Again, litter type effects were not additive, i.e. seedling height per pot in the mixed litter treatment lay significantly above the mean of seedling height in grass and ragwort litter (p = 0.001, Fig 3). Contrast

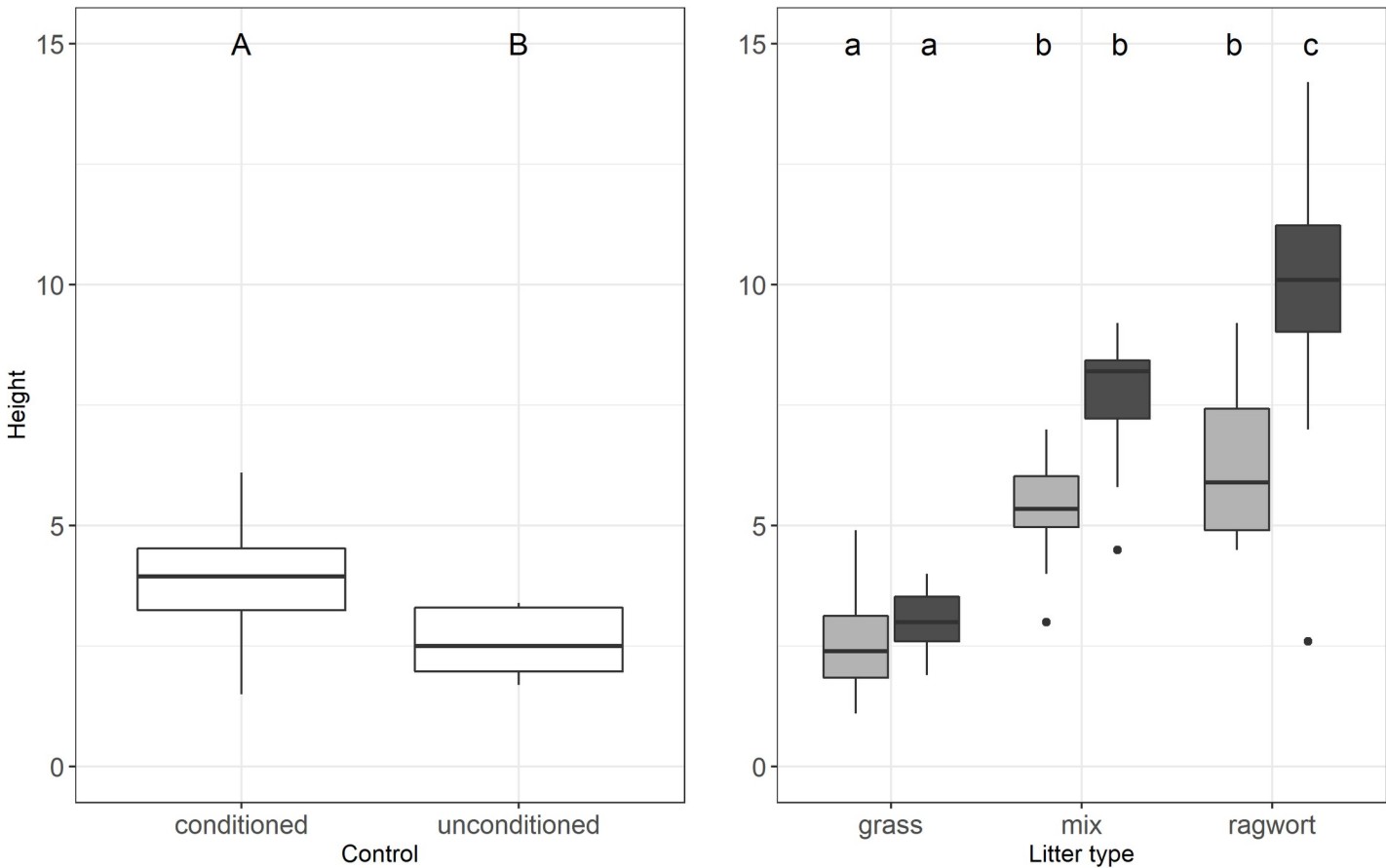

**Fig 3. Seedling height—Effects of soil conditioning on the control and the two-way interaction of litter type and litter amount.** In the box plots middle lines represent median, boxes represent the first and third quartiles, lower and upper bars represent the minimum and the maximum and points represent outliers (i.e. points above 1.5 SD). Grey: 200 g/m² litter, black: 400 g/m² litter. Lower case letters indicate significant interactions between treatments (TukeyHSD, P≤ 0.05).

analyses of the control against the three litter types revealed no significant difference of height between the control and grass litter (p = 0.39) but significant differences between control and litter mixture as well as ragwort litter (both p ≤ 0.0001).

In contrast, litter amount had a much smaller, still significant effect, i.e. higher litter amounts led to taller seedlings (Fig 3). Seedling height is the only response variable where the control was significantly influenced by soil type, i.e. ragwort conditioned soil led to significantly taller seedlings (3.9 ± 0.5 cm vs. 2.6 ± 0.3 cm).

The mean leaf number of plants growing in ragwort litter was highest compared to in plants growing in grass litter and the combination of both litter types (ragwort 3.59 ± 0.12, grass 3.15 ± 0.05, mixture 3.31 ± 0.08). Analyses of variance revealed that mean leaf number was significantly affected by litter type and amount but not their interaction (Table 1 and Fig 4). The variance explained by litter amount was twice as high as that explained by litter type (Table 1). Thus, mean leaf number is the only measured variable that seems to be stronger influenced from litter amount than litter type. Effects of litter type were additive, i.e. the leaf number in the litter mixture did not differ significantly from the expected mean effect of pure grass and ragwort litter (p = 0.49, Fig 4). Plants growing in high litter amounts (400 g/m²) had significantly more leaves per plant than plants growing in the low litter amounts (200 g/m²). Contrast

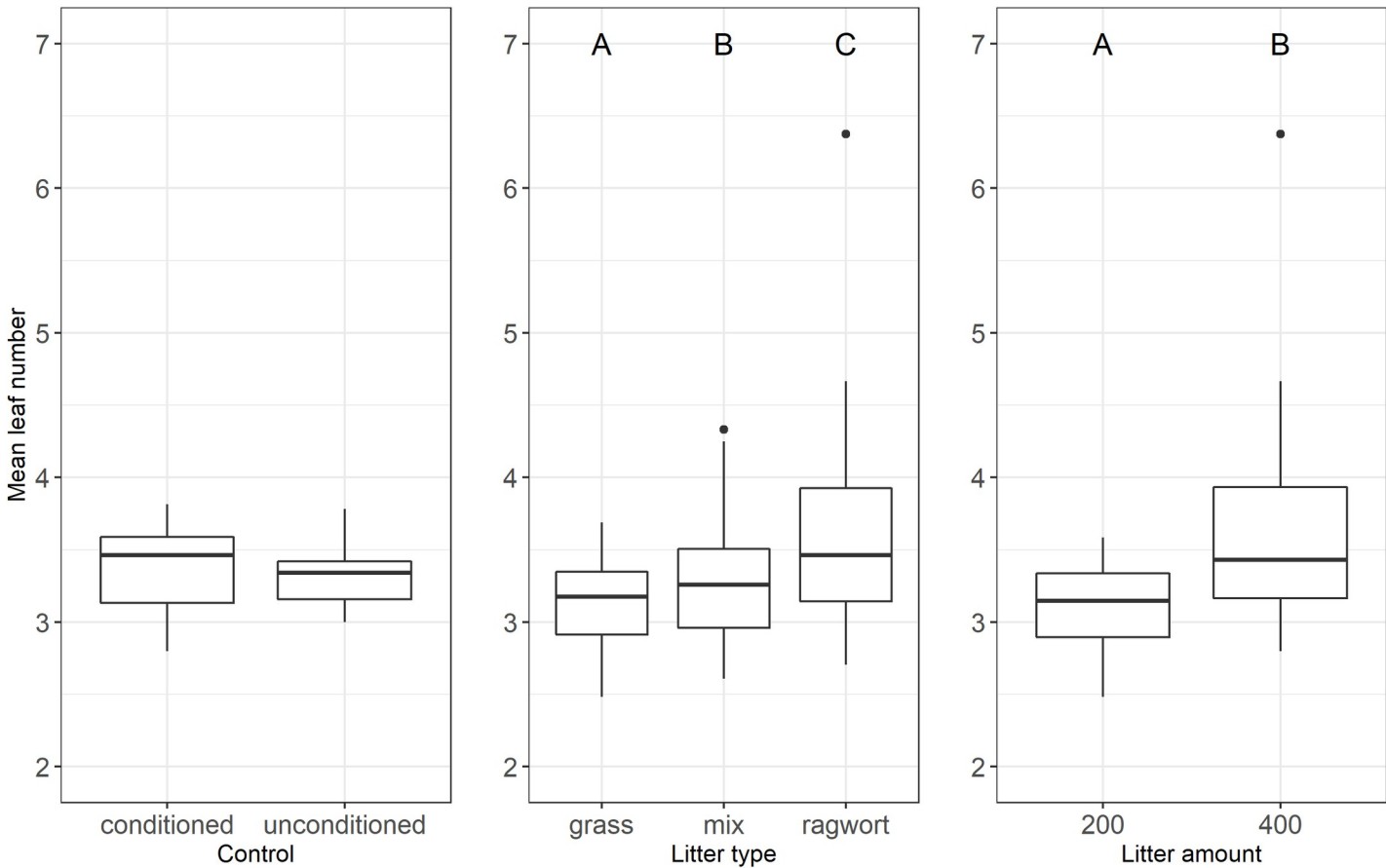

**Fig 4. Mean leaf number under soil conditioning in the control (left), different litter types (grass, mix, ragwort (middle)) and different litter amounts (200 g/m² litter, 400 g/m² litter (right)).** In the box plots middle lines represent median, boxes represent the first and third quartiles, lower and upper bars represent the minimum and the maximum and points represent outliers (i.e. points above 1.5 SD). Upper case letters indicate significant main effects between treatments (TukeyHSD, P≤ 0.05).

analyses revealed no significant difference in leaf number between the plants growing in the control and in any of the three litter types (p > 0.05).

## Discussion

Our experimental results suggest that litter type as well as litter amount do influence seedling emergence and growth, whereas conditioned soil did not when combined with litter treatments. Our main finding is, that although ragwort litter reduced the number of conspecific emerging seeds, seedlings actually performed better in ragwort litter showing increased biomass, height and leaf numbers. Therefore, ragwort litter cannot be used to control reduce ragwort abundance.

Litter type accounted for a much higher portion of the variance in seedling number, biomass and height than litter amount. In contrast, other studies reported that litter amount was more relevant than litter type [47, 48]. This highlights the species-specific nature of litter effects [17]. One explanation for this pattern could be that ragwort litter is indeed autotoxic and induces self-thinning by negatively influencing the emergence of conspecific seedlings [11]. Other possible factors are impacts of different litter types on microclimate, the rate of

decomposition and nutrient release or the relative difference between litter amounts compared to differences between litter types.

Seedling emergence was highest in grass litter, lowest in ragwort litter and intermediate in the litter mixture. Germination was highest in medium compared to no (control) or high litter amounts. In contrast, all parameters measuring seedling growth (biomass, height, mean leaf number) were highest in high amounts of litter and/or ragwort litter. The positive effect of grass litter and medium litter amounts on seedling emergence might be due to physical litter effects like preserving a relatively constant temperature and soil moisture [49]. In our experiment, the same amount of grass litter covered a larger proportion of the soil surface than ragwort or mixed litter (high amount: grass 90 ± 3%, mixed 85 ± 4% and ragwort 67 ± 8%, low amount: grass: 85 ± 3%, mixed 70 ± 9%, ragwort 37 ± 5%). In accordance with a study by Eckstein and Donath [49], which revealed a rise in soil moisture when litter cover increased, we assume that seedlings experienced more favourable water availability in grass litter and medium litter amounts. In high litter amounts however, negative effects of litter prevailed over positive effects. Laboratory experiments of van der Meijden et al. [15] demonstrated that seed germination of ragwort increased with higher soil humidity. We also observed that ragwort litter decayed faster than grass litter (personal observation). This is in line with general findings that litter of forb species such as ragwort tends to decay faster than grass litter [50]. Schuster et al. [51] found that the mass of ragwort litter remaining after 30 days was only half of the amount of grass (*Alopecurus pratensis*) litter. Since the protective properties of a litter cover against desiccation and temperature extremes increases with litter amount present, seedlings emerging from beneath faster decaying ragwort litter might have experienced less protection than those seedlings emerging beneath grass litter [14, 49]. Thus, the negative effect of ragwort litter on ragwort seedlings emergence may be a result of fewer physical benefits seedlings experienced from ragwort litter compared to grass litter. In addition, effects of leachate may depend on the specific decomposition rate of different litter types. As leachates released from the litter are declining very rapidly after senescence [52], intensity of seedlings' exposure to leachate depends on the decomposition rate of the litter present. Thus, ragwort seeds and seedlings emerging from ragwort litter, which decays fast [51], were only exposed to the ragwort leachate for a short time. Consequently, we assume that leachate effects on seedling emergence were more pronounced than on seedling growth.

In general, the amelioration of the physical environment by litter (e. g. on soil moisture), rather than its chemical composition, seems to be the main factor boosting seedling growth in short time scales [20]. However, the size of chemical autotoxic effects and therefore their relative importance in relation to physical and mechanical or other biotic effects are species-specific [53]. Several studies show the occurrence of autotoxicity of ragwort [11, 34], which suggests that chemical effects are an additional mechanism for lower seedling emergence from beneath ragwort litter. Van de Voorde [34] reported decreasing germination with increasing concentrations of ragwort extract. As ragwort produces many seeds with high germination potential and low dispersal ability [1, 40, 41] autotoxic effects may avoid intraspecific competition in ragwort and thus may lead to lower seedling numbers but increased seedling vitality.

In contrast of Van de Voordes notion of a negative soil-feedback in ragwort [34], we found only weak indications of effects of ragwort conditioned soil on seedling emergence and growth (Table 1). In fact, the only significant effect of conditioned soil occurred on seedling height in controls, i.e. seedlings in conditioned soil grew bigger than in unconditioned soil. This finding is in contrast to findings of van de Voorde [32], who reported autotoxic effects of ragwort on its seedling development. However, on average the observed difference in plant height between ragwort seedlings growing in conditioned versus those growing in unconditioned soil including control and litter addition treatments was only 2 mm (less than 4% of the average height of about 54.5 mm). Therefore, we expect that this difference is too small to be of ecological

relevance. In other studies, looking at several seedling cohorts, those cohorts usually differed several centimeters in height [54–56].

Mixed litter resulted in fewer emerged seedlings than in grass litter, but more emerged seedlings compared to ragwort litter, which suggests that the effects of grass and ragwort litter simply added up to a neutral effect. This supports the mass-ratio hypothesis by Grime [21] for the factors seedling number, biomass per plant and mean leaf number, i.e. mixed litter effects did not deviate positively nor negatively from the expected combined effects. However, for biomass per pot and plant height, effects in mixed litter were higher than could be assumed from the simple addition of grass and ragwort. Thus, ragwort growth profited in single aspects from mixed litter layers. Thus, from a management point of view, leaving litter on ragwort infested sites does slightly facilitate ragwort growth when biomass of ragwort and grass is balanced. However, if ragwort is dominating the standing biomass, germination of its seedlings might indeed be limited. When litter cover is too high, the soil is shaded and seedling emergence gets lower. Accordingly, compared to the control treatment, seedling emergence was lowest from beneath 400 g of litter and not reduced from beneath 200 g of litter cover. The reduced seedling emergence of ragwort from beneath 400 g of litter is very likely linked to the relatively small seed mass of ragwort (0.22 mg; [57]). The smaller the seeds are, the less resources a seed can spend to successfully penetrate through a litter cover [58]. In addition, small seeded species, like ragwort, tend to depend on higher light levels for germination than large seeded species [59, 60]. Since photosynthetic active radiation (PAR) decreases with increasing litter cover [49] not only will germination be lower, but germinated seedlings will also lack light resources for successful penetration of the litter cover. This is also in accordance with van der Meijden and van der Waals-Kooi [15], who found that emergence in ragwort significantly decreased when seeds were placed beneath a sand cover of more than 4 mm. Furthermore, Facelli et al. [61] showed that ruderal species, such as ragwort, tend to be more sensitive and thus more likely to be negatively affected by high amounts of litter. Therefore, ragwort seedlings might be reduced by a mulching regime on productive sites where litter amounts are medium to high.

In contrast to the findings for seedling emergence, seedlings in ragwort litter built up more biomass, grew higher and tended to have more leaves than seedlings in grass litter. Biomass and height are the two variables where we found a significant interaction between litter type and litter amount. This could be explained by a reduction in competition for the few seedlings that emerged in ragwort litter. The interaction was quantitative and weak compared with the effects of litter type and amount alone (Table 1). Seedlings emerging from beneath high amounts of litter and/or ragwort litter built up the highest biomass and got tallest. This is also diametrical to the findings of van de Voorde et al. [34], who reported lower biomass for seedlings in ragwort influenced environments. These conflicting results could be due to different methodology. While van de Voorde et al. [34] harvested seedling biomass after 19 days, seedlings in our study were harvested about 50 days later. While van de Voorde's results were supported by our early observations (seedlings looked smaller and thinner early during early development in ragwort litter), seedlings started to grow bigger after ragwort litter had decomposed. Thus, the longer duration of our experiment compared to the experiment by van de Voorde et al. [35] allowed us to observe a change in the importance of litter effects in relation to intraspecific density dependent competitive effects [49], which is further supported by the fact that lower seedling numbers from beneath grass litter resulted in seedlings of higher biomass and height. Loydi et al. [47] already identified, that while litter plays a major role for seedling emergence, for instance, via phytotoxins, competition is more important for seedling development later during seedling growth. Significantly increased mean leaf numbers in seedlings in high litter amounts and in ragwort litter compared to plants in grass or mixed litter

might be linked to lower intraspecific competition as there were fewer seedling numbers in the ragwort pots or by less physical hampering compared to grass litter [58]. However, indirect positive effects of reduced seedling emergence in boosting seedling biomass could be especially pronounced in situations where, as in the present experiment, intraspecific competition prevails and may overlay different environmental constrains under field conditions [13, 62, 63].

Another explanation for superior performance of emerged seedlings in high litter amounts and ragwort litter might be an improved availability of nutrients [12], as more biomass contains more nutrients and ragwort litter decomposed quite fast. This is supported by our observation that even though the number of seedlings in the control pot and in the ragwort litter pot with 200 g/m$^2$ was about the same and thus competition among seedlings was comparable, biomass of seedlings growing in ragwort litter pots was still higher. However, it is questionable if nutrients from decaying litter are converted to plant available nutrients within 70 days. Effects of different litter amounts on nutrient availability are generally relatively low [47]. Another reason for lighter and smaller seedlings in lower amounts of litter could be due to the missing investment of seedlings in higher growth when they have already reached light [64]. Siebenkäs et al. [65] compared seedling growth in the same developmental stage as seedlings in our experiment and found that plants tend to grow higher when there is more shade. Thus, seedlings may grow higher in higher litter amounts to avoid its shade and in order to reach light. Renne et al. (2006) found that small differences in height of seedlings early on can produce large differences in their final biomass [66] indicating that early shading can translate in higher heights, even if shade is gone.

## Conclusion

We found both negative and positive effects of ragwort litter on conspecific seedling emergence and growth. While negative effects can be linked to physical and autotoxic chemical ragwort litter effects in the stage of seedling emergence, those seedlings that emerged successfully may have profited from a lower intraspecific competition and may had more resources to invest than seedlings that emerged beneath grass or mixed litter. Due to the ambivalent nature of our results, these findings cannot easily be translated into management recommendations for ragwort-infested sites. However, the effects of mixed litter on ragwort recruitment were either neutral or even promoted ragwort growth. Therefore, when we assume that biomass on mowed ragwort pastures consists roughly of similar amounts of grass and ragwort, we expect that leaving litter containing ragwort biomass on the grassland will have no negative effect on ragwort population density or growth. As already our results for a simple controlled experiment do not lead to a clear hint of the effectiveness of ragwort litter in reducing ragwort population density, we do not expect pronounced effects in an experiment under more natural conditions.

## Supporting information

**S1 Table. Raw data for seedling emergence and seedling growth according to different treatments.**
(CSV)

**S2 Table. Results of a three-way ANOVA on the effects of litter type, litter amount and soil on ln biomass per plant.**
(DOCX)

**S1 Fig. Biomass per plant—effects of the two-way interaction of litter type and litter amount.** In the box plots middle lines represent median, boxes represent the first and third quartiles, lower and upper bars represent the minimum and the maximum and points represent outliers (i.e. points above 1.5 SD). Grey: 200 g/m$^2$ litter, black: 400 g/m$^2$ litter. Lower case

letters indicate significant interactions between treatments (TukeyHSD, P$\leq$ 0.05).
(DOCX)

## Acknowledgments

We thank Astrid Dempfle for her support in statistical analyses. Furthermore, we thank Hollyn Hartlep for her valuable help in proofreading.

## Author Contributions

**Conceptualization:** Henrike Möhler, Geeltje Marie Bauer, Tobias W. Donath.

**Formal analysis:** Henrike Möhler, Geeltje Marie Bauer, Tobias W. Donath.

**Funding acquisition:** Tim Diekötter, Tobias W. Donath.

**Investigation:** Geeltje Marie Bauer.

**Methodology:** Henrike Möhler, Geeltje Marie Bauer, Tobias W. Donath.

**Project administration:** Henrike Möhler, Tobias W. Donath.

**Resources:** Tim Diekötter, Tobias W. Donath.

**Supervision:** Henrike Möhler, Tobias W. Donath.

**Validation:** Henrike Möhler, Tobias W. Donath.

**Visualization:** Henrike Möhler, Geeltje Marie Bauer, Tobias W. Donath.

**Writing – original draft:** Henrike Möhler, Geeltje Marie Bauer.

**Writing – review & editing:** Henrike Möhler, Tim Diekötter, Tobias W. Donath.

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
