## [Decision Letter · Decision Letter 0]

24 Aug 2020

PONE-D-20-12615

Conspecific and heterospecific litter effects on seedling emergence and establishment in ragwort (Jacobaea vulgaris Gaertn.)

PLOS ONE

Dear Dr. Möhler,

Thank you for submitting your manuscript to PLOS ONE. After careful consideration, we feel that it has merit but does not fully meet PLOS ONE’s publication criteria as it currently stands. Therefore, we invite you to submit a revised version of the manuscript that addresses the points raised during the review process.

We look forward to receiving your revised manuscript.

Kind regards,

Craig Eliot Coleman, PhD

Academic Editor

PLOS ONE

Journal Requirements:

2. Please amend either the title on the online submission form (via Edit Submission) or the title in the manuscript so that they are identical.

Reviewers' comments:

Reviewer's Responses to Questions

**Comments to the Author**

1. Is the manuscript technically sound, and do the data support the conclusions?

Reviewer #1: Yes

Reviewer #2: Partly

2. Has the statistical analysis been performed appropriately and rigorously? 

Reviewer #1: Yes

Reviewer #2: Yes

3. Have the authors made all data underlying the findings in their manuscript fully available?

Reviewer #1: Yes

Reviewer #2: Yes

4. Is the manuscript presented in an intelligible fashion and written in standard English?

Reviewer #1: Yes

Reviewer #2: Yes

5. Review Comments to the Author

Reviewer #1: This manuscript describes a container experiment in which germination and seedling growth (10 weeks) of ragwort was tested in a factorial experiment with litter type, litter quantity, and soil conditioning as treatments. The study’s rationale is the challenge that ragwort is a significant management problem in rangelands, so understanding strategies (including cutting, etc.) to reduce ragwort cover is important.

The manuscript is well-organized and well-written, but for some minor points of clarity that I noted in my marked-up PDF. The study design was appropriate and the analysis sound. The conclusions, that litter type and quantity had a strong negative effect on seedling number and a positive effect on biomass per plant. Many fewer seedlings emerged in pots with ragwort litter (as opposed to grass or mixed litter), though the plants that did emerge were larger.

The authors missed an opportunity to synthesize these observations by considering the per pot biomass of ragwort. The larger per-plant biomass of under ragwort litter is not inconsistent with a negative effect of litter on germination and early-plant growth. I expect that the total biomass – the sum of individual plants’ biomass per pot – was lower in ragwort litter pots. That would offer evidence that ragwort self-limits itself and perhaps would offer a clearer management recommendation. “Self limitation” could be a broader concept that this study tests, if the authors choose. More importantly, the conclusion could change, as this finding would suggest a specific strategy for managers hoping to limit ragwort.

The Discussion should be reorganized to be clearer and shorter. The division into headings based on the main treatments – litter type, litter quantity, and soil conditioning – prevented a synthetic consideration of the comparisons. I suggest that the Discussion more clearly compare the relative effects of each at the start of the Discussion so that the reader clearly gains a synthetic picture. Then, the relative importance of litter type, litter quantity, and soil conditioning can be compared to other studies – many of which have also combined one or more of your same treatments (though not all three).

Specific comments:

- Throughout your headings and text, you should phrase your measurement of ragwort “biomass” or “growth”, rather than “establishment.

- Line 125 – Cite the taxonomic authority as you did in the title. In fact, you can remove it from the title and just have it here.

- Line 138 – State that the mixture was 50:50.

- Lines 172-175 – Was biomass harvested after ~6 weeks, not 10? Why?

- Lines 178-184 – I would like to see more comparison of these control treatments. Did they differ? Is there evidence that the conditioned soil had more seedlings? What is the justification for using the MEAN of these two treatments as your LnRR.

- Results – Please emphasize more the direction of the effect primarily, and the “significance” secondarily. The reader only late in most paragraphs learns whether the effect of a treatment decreased ragwort, yet that is the key biological question.

- Why are the figures set up so differently? I would like to see more consistency.

- Line 310-313 - Doesn’t the (mostly) absence of an effect of soil conditioning refute that likelihood of allelochemical effect?

I have also included a copy of the manuscript with my comments.

Signed,

Dr. Jeffrey D. Corbin

Union College

Reviewer #2: Comments

In this manuscript Möhler et al. present findings from a mesocosm experiment examining the impacts of soil conditioning, litter species, and litter amount on the germination and initial establishment of invasive Jacobaea vulgaris (ragwort). They find that ragwort litter suppressed germination, but that resulting germinants grew larger than germinants arising from controls or from soil covered with grass litter. The authors ask valuable questions; particularly since experiments combining soil conditioning and litter impacts are relatively scarce. Overall, the manuscript has merit and is worth publishing, but will require significant revision beforehand. In particular, I recommend that the analyses be revised and expanded (see below). These are major revisions, but may also warrant a reject decision.

Perhaps my largest concern with the manuscript is that it is over-eager to draw conclusions outside the scope of the design. There is a fair amount of attention given to litter leachates as a possible inhibitory factor, but this wasn’t tested and although likely, is not necessarily the sole explanation for the observed results. More importantly, the manuscript concludes that leaving ragwort litter in the field has no net effect on ragwort establishment, but this is not adequately assessed by the presented analyses. To better understand this, the authors should evaluate whether grass-ragwort mixtures had an effect that deviated from the expected value based on those two litters in monoculture. Based on Fig 3, it appears that ragwort litter (which is different than green biomass – something that must be kept in mind) disproportionately stimulates germinant growth. If so, it would mean that incorporation of ragwort litter into mixtures with grass litter would result in more favorable conditions for germinants and that ragwort litter could facilitate a positive feedback loop favoring more ragwort. Managers should then seek to limit ragwort litter as much as possible (contrary to the current conclusion).

The analyses are structured around response ratios. Most of the results and discussion focus on differences in observed values and not on deviation from control values. So, there is an inherent disconnect between how data were analyzed and presented, and how they are discussed. This is confusing, and not particularly robust for most comparisons because it ignores variance of control units. When attempting to discern if litter mixtures behaved additively, the response ratio analyses make sense (although no such analysis is actually presented here – a missed opportunity since mixed litter studies are also uncommon). In all other cases, using a simpler factorial design would be more accurate and more easily interpreted since it A) preserves variance of the controls, and B) directly reflects observed values. The structure of the analyses should be reconsidered.

The results are vague. It is not sufficient to only state that levels differed. Instead, the direction and magnitude of the change should also be presented in text (e.g. seedlings grew X% taller under ragwort litter than under grass litter). The results could also be clarified by tying statements of significance to specific tests in Table 1. As is, the table is presented in passing and it is unclear exactly which tests correspond to any given statement of significance.

The introduction lacks detail and does not adequately establish the mechanisms tested. More detail is needed to address the specifics of how mechanisms are hypothesized to function. Stating only that litter facilitated a change in abiotic conditions is insufficient. Instead, it must be made clear how conditions changed (resource, direction, magnitude) and what that effect occurred.

The introduction does not sufficiently support the hypotheses. Most of the introduction focuses on the question of how litter presence affects seedling establishment (vaguely described). However, the rationale behind testing litter amount and soil conditioning, as well as testing multiple species, is unclear. Each of these contrasts must be explicitly described in the introduction.

The manuscript requires minor editing for English.

Specific comments

31. provide units in square meters

48. “poisonous” requires some context – it is toxic to animals if consumed. This is important since “poisonous” is also used to describe things like Toxicodendron

51. you mean that it is challenging to raise cattle if ragwort is abundant?

56. citation required

58. If the concern is that cattle will consume the biomass (either in the form of hay or as living plants), wouldn’t leaving litter in place also be risky to cattle?

63. Litter affects

64. here and throughout, be more specific. “changing” is too vague and does not illustrate the mechanism.

66. The mechanisms have not been sufficiently described. More detail is needed in how and why litter affects microclimate and nutrient availability, either within the context of ragwort specifically or herbaceous weeds in general.

68. paragraph needs a topic sentence and clearer structure. We jump from addressing seedling establishment directly, to abiotic impacts of litter, to litter mixing effects. Break these concepts up into separate paragraphs and provide clear transition between each.

68. which type of seedlings?

76. it is confusing to mention phytotoxins in a section where you are predominantly describing physical impacts of litter

83. Is it intuitive? What about allee effectS?

84. assumed by whom?

88. Is it just this one study or were there more?

89. Incomplete sentence – what did they show about litter and soil?

90. What is the feedback? Is it intra-specific or inter-specific? Provide detail.

90. “differences” is too vague

91. is “probably driven by differences in the community” your speculation or that of the cited works?

92. Negative effects on which response?

95. the introduction should establish which factors are important to seedling establishment and seedling emergence as well as highlight differences between the two. Currently no support for this hypothesis is presented.

103. In general, this section should not be necessary. It should be enough to state the hypotheses or questions without going back and explaining each. At the point that the hypotheses are presented, the reader should already be familiar enough with the mechanisms to understand the hypotheses without explanation. The current format of presenting a question and then detailing the predicted response is confusing and seems to be used instead of adequate background information beforehand. Furthermore, there are no citations presented within these explanations, so it is unclear what is hypothesis and what is already known.

107. What do you mean by “congruent” – that there will be no difference between litter types?

114. Confusing. Break into multiple sentences.

124. This information would be more useful in the Introduction since the paper focuses on ragwort specifically. Also, additional information on the ecological effects of ragwort would be helpful.

144. Schleswig-Holstein, Germany

149. Provide GPS coordinates

151. The soil is still conditioned, just not by ragwort. It would be more accurate to describe it as “grass-conditioned” rather than “unconditioned”, assuming that the dominant species where soil was collected were grasses.

153. That is a relatively low-intensity drying treatment. Did you verify that litter had reached constant mass by that point?

154. How do these litter treatments correspond to natural conditions?

155. This is a weakness of the experimental design that the authors must address. The study focuses on specific litter effects, and while ragwort is maintained as a specific treatment, the grass litter treatment is poorly controlled. The exact composition of the grass litter treatment is unclear. Ideally, this would have been a single grass species, but instead appears to be a presumably variable mixture of different species. This severely limits the authors’ ability to detect litter mixing effects since the baseline additive condition would vary with the composition of the grass litter. This is true for all litter experiments to some extent since litter quality is variable even within a single plant, but by including multiple grass species as a single treatment, this error is enhanced.

158. Where are data from these pots presented, or how were they considered in the analyses? If litter type and amount affects germination, wouldn’t you also need to expose control pots to the same conditions to accurately account for seedbank under the considered conditions?

165. When was litter applied?

214. This is a 56% increase, not a 33% increase. This is also somewhat confusing because your analyses were using LnRR and not on the actual observed data. So, when we look at Fig 1, we do not see a 56% increase, we only see that ragwort litter suppressed emergence and grass stimulated emergence.

215. “interestingly” is not needed

216. Unclear. What do you mean by “opposite?” Where is this shown? Was there a species by amount interaction – Fig 1 shows no effect of 2 g/dm2 of litter.

217. There is an over-reliance on “i.e.” throughout the manuscript. Be concise.

218. Also worth noting that 2 g/dm2 had no impact.

219. MSQ is not necessary since you have provided F statistics

236. all interactions are quantitative – overly wordy. Erroneous bold.

238. Fig 2 does not support this since grass litter error bars overlap and do not differ from 0 (controls).

241. This appears to be redundant with line 239.

255. new paragraph

259. Quantify change. This is an issue throughout the results.

289. But you also only tested 2 levels of litter. How well these levels correspond to real world conditons as well as the conditions tested by those other studies is an important consideration.

290. You did not test for autotoxicity and cannot directly support this claim. There are other factors that could explain the presented trends.

293. Although lower, ragwort would also have impacts on microclimate.

296. Given that litter did not consistently cover the soil surface, does that mean that using mass per area is actually a fair method for comparing litter types? Type is confounded by cover (even if mass per area is constant).

304. which grass litter? Be specific.

308. But ragwort had lower germination than controls, which had no litter at all. So, even though ragwort decayed faster than grass litter, it should still have a neutral-positive impact relative to controls.

311. Over short time scales, yes, although this was not tested.

315. remove “inline”

322. Not exactly. Even if grasses and ragwort interacted, we would generally expect an intermediate response. What matters is if that response differs from the average of grasses and ragwort (assuming a 1:1 mixture grass:ragwort). The available data could easily test this.

326. That depends on the cover of ragwort (and on whether there was a mixing effect). If there is a perfect mixture of grasses and ragwort, your results suggest a neutral effect. If ragwort is abundant, then there would be a negative impact of leaving litter on ragwort germination. Also, is ragwort litter commonly removed by burning or raking? It is important to note that litter has different physical and chemical properties than green biomass, so this isn’t exactly comparable to land manager mowing and leaving the trimmings in field.

341. If the biomass result is primarily a consequence of competition, is it fair to ascribe it to the litter type in the first place?

349. Yes, but it takes time for these nutrients to be converted into forms usable by plants

414. See comments above – this is not a robust conclusion.

Fig 1. Increase axis label size. Why are groups using upper case letters here and lower case letters in Fig 2? Unify if not intentional.

Fig 2. Add legend to figure. Increase axis label size.

Fig 3. Increase axis label size.

Figs 1,3,4 – It would be helpful to have all of these presented in the format of Fig 3 with type, amount, and soil presented in Fig 1 and 4 as well. Even though soil was not significant in figs 1 and 4, it would provide consistency across a somewhat complex set of analyses and would make comparisons easier for the reader.

6. PLOS authors have the option to publish the peer review history of their article (what does this mean?). If published, this will include your full peer review and any attached files.

Reviewer #1: **Yes: **Jeffrey D Corbin

Reviewer #2: No

---

## [Author Response · Author response to Decision Letter 0]

19 Nov 2020

Dear Editor and Reviewers,

We thank you very much for your time, efforts and helpful comments and critiques. Based on both reviews we created an updated and improved manuscript.

The major changes we made were first, reanalyzing all the data with real observed data instead of derived Log response ratios. Second, adding a comparison between data for “mono” and mixed litter types. We also tried to give a more synthetic picture of the results and their context in findings of others in the discussion, which we hope is now shorter and clearer.

Unfortunately, we must disagree on some points, listed below. Further down, we would like to directly address these points, with your critique stated repeated (following R1/R2) and our answer (following A). Minor points were track changed directly in the manuscript.

In any case, we thank you for taking our manuscript into consideration anew and thank you again for your helpful critique and comments.

Yours sincerely

Henrike Möhler

Reviewer 1:

R1: The authors missed an opportunity to synthesize these observations by considering the per pot biomass of ragwort. The larger per-plant biomass of under ragwort litter is not inconsistent with a negative effect of litter on germination and early-plant growth. I expect that the total biomass – the sum of individual plants’ biomass per pot – was lower in ragwort litter pots. That would offer evidence that ragwort self-limits itself and perhaps would offer a clearer management recommendation. “Self limitation” could be a broader concept that this study tests, if the authors choose. More importantly, the conclusion could change, as this finding would suggest a specific strategy for managers hoping to limit ragwort.

A: We added information on per pot biomass throughout the manuscript. However, as the results of per plant and per pot biomass do not differ in their general pattern (see figure below or figure added in supporting information), we cannot support reviewer 1’s guess, that biomass was lower in ragwort litter pots and thus do not further discuss self-limitation. However, we did discuss self-thinning in the manuscript (line 324, 363, 415ff).

R1: The Discussion should be reorganized to be clearer and shorter. The division into headings based on the main treatments – litter type, litter quantity, and soil conditioning – prevented a synthetic consideration of the comparisons. I suggest that the Discussion more clearly compare the relative effects of each at the start of the Discussion so that the reader clearly gains a synthetic picture.

A: We reorganized the discussion accordingly and refrained from the division in headings based on the main treatments.

R1: Throughout your headings and text, you should phrase your measurement of ragwort “biomass” or “growth”, rather than “establishment.

A: We followed this advice.

R1: Line 125 – Cite the taxonomic authority as you did in the title. In fact, you can remove it from the title and just have it here.

A: We followed this advice.

R1: Line 138 – State that the mixture was 50:50.

A: We followed this advice.

R1: Lines 172-175 – Was biomass harvested after ~6 weeks, not 10? Why?

A: We corrected the wording. It was indeed 10 rather than 6 weeks.

R1: Lines 178-184 – I would like to see more comparison of these control treatments. Did they differ? Is there evidence that the conditioned soil had more seedlings? What is the justification for using the MEAN of these two treatments as your LnRR.

A: Control treatments did not differ unless for height. We added graphs of how the controls behaved for each parameter discussed. 

R1: Results – Please emphasize more the direction of the effect primarily, and the “significance” secondarily. The reader only late in most paragraphs learns whether the effect of a treatment decreased ragwort, yet that is the key biological question.

A: We agree and followed this advice.

R1: Why are the figures set up so differently? I would like to see more consistency.

A: We followed this advice.

R1: Line 310-313 - Doesn’t the (mostly) absence of an effect of soil conditioning refute that likelihood of allelochemical effect?

A: Allelochemical effects can act on very short time scales and not translate in soil effects. Thus, absence of soil conditioning and presence of autotoxicty are not contradictory. 

R1: Abstract needs synthesis.

A: We followed this advice.

R1: Line 104-121 Hypotheses are clear. Nice job.

A: Thanks for this comment.

Reviewer 2:

R2: Perhaps my largest concern with the manuscript is that it is over-eager to draw conclusions outside the scope of the design. There is a fair amount of attention given to litter leachates as a possible inhibitory factor, but this wasn’t tested and although likely, is not necessarily the sole explanation for the observed results. 

A: We do not quite understand this claim. Maybe we might have not described the design clear enough, thus we rephrased and added information on this in the manuscript now. We exposed seedlings of ragwort to ragwort litter and left the litter for decay under outdoor conditions. Thus, seeds were exposed to litter leachates that would also occur under natural conditions. It is certainly true that we did not only assess the chemical effects induced by leachate but also physical and mechanical litter effects. Therefore, we also discuss these in detail in the discussion section.

R2: More importantly, the manuscript concludes that leaving ragwort litter in the field has no net effect on ragwort establishment, but this is not adequately assessed by the presented analyses. To better understand this, the authors should evaluate whether grass-ragwort mixtures had an effect that deviated from the expected value based on those two litters in monoculture. Based on Fig 3, it appears that ragwort litter (which is different than green biomass – something that must be kept in mind) disproportionately stimulates germinant growth. If so, it would mean that incorporation of ragwort litter into mixtures with grass litter would result in more favorable conditions for germinants and that ragwort litter could facilitate a positive feedback loop favoring more ragwort. Managers should then seek to limit ragwort litter as much as possible (contrary to the current conclusion).

A: We added analysis on whether effects were purely additive or not. Indeed, concerning the two variables height and biomass per pot effects deviated positively, e.g. litter mix facilitated higher growth and led to higher biomass per pot. However, there was an only additive effect in biomass per plant and mean number of leaves. Furthermore, even though seedling vigor was positively influenced, seedling emergence was not. Thus, deriving a net effect favoring more ragwort cannot be concluded. Because of these ambivalent results we conclude that leaving or removing litter on the grassland will not substantially influence ragwort density. We clarified this in our conclusions.

R2: The analyses are structured around response ratios. Most of the results and discussion focus on differences in observed values and not on deviation from control values. So, there is an inherent disconnect between how data were analyzed and presented, and how they are discussed. This is confusing, and not particularly robust for most comparisons because it ignores variance of control units.

A: Log response ratios are a nice and valid approach to handle hanging control groups (in the current study the control pots (no litter) acted at the same time as the control for grass litter, grass-senecio-mix and pure senecio litter), but they come at the expense of some drawbacks; one (i.e. ignoring variance of control units) was highlighted by you and since we follow your argument we decided to rearrange the analyses. We changed the analyses and worked with the original to data make it easier to assess direct changes in factors like seedling number, biomass, height, and mean leaf number. In this approach, we did not include the controls in the complete ANOVA-model. We were interested in the differential effects of the three litter types; and not in the general differences of no litter cover vs. litter cover present (therefore, we did also not refer to a control litter treatment in our research question/expectations section). 

Still we included a control in the experimental set up for informational reasons and tried to include this data in our original analyses via the log response approach – at the cost of the above-mentioned drawback. Based on the aim of our study and our research questions we could have also just skipped the controls completely from the experiment/manuscript but since they might be some merit reporting also in the updated manuscript we both included the controls in the graphs as well as doing an analyses control vs. litter treatments and analyzed for soil effects between the control treatments. 

R2: When attempting to discern if litter mixtures behaved additively, the response ratio analyses make sense (although no such analysis is actually presented here – a missed opportunity since mixed litter studies are also uncommon). 

A: We added analysis on whether effects were additive or not. To this end, we used contrasts to specifically test whether the effect of mixed litter on response variables was mid-way between that of grass and ragwort alone (additive effect of pure type litters in the mixture).

R2: In all other cases, using a simpler factorial design would be more accurate and more easily interpreted since it A) preserves variance of the controls, and B) directly reflects observed values. The structure of the analyses should be reconsidered.

A: We adjusted or analyses to observed values to make it easier to see direct changes in factors like seedling number, biomass, height, and leaf number. Controls are displayed next to the treatments we are interested in. 

R2: The results are vague. It is not sufficient to only state that levels differed. Instead, the direction and magnitude of the change should also be presented in text (e.g. seedlings grew X% taller under ragwort litter than under grass litter). The results could also be clarified by tying statements of significance to specific tests in Table 1. As is, the table is presented in passing and it is unclear exactly which tests correspond to any given statement of significance.

A: We added information on direction and magnitude of change in the text and tied them to Table1.

R2: The introduction lacks detail and does not adequately establish the mechanisms tested. More detail is needed to address the specifics of how mechanisms are hypothesized to function. Stating only that litter facilitated a change in abiotic conditions is insufficient. Instead, it must be made clear how conditions changed (resource, direction, magnitude) and what that effect occurred.

A: We added information on the mechanisms.

R2: The introduction does not sufficiently support the hypotheses. Most of the introduction focuses on the question of how litter presence affects seedling establishment (vaguely described). However, the rationale behind testing litter amount and soil conditioning, as well as testing multiple species, is unclear. Each of these contrasts must be explicitly described in the introduction.

A: We cannot confirm this critique as we explicitly described effects of litter mix (line 90ff), litter amount (line 90ff) and conditioned soil (line: 102ff). However, we tried to better carve out those points.

R2 31. provide units in square meters

A: done

R2 48. “poisonous” requires some context – it is toxic to animals if consumed. This is important since “poisonous” is also used to describe things like Toxicodendron

A: We rephrased and avoided the word “poisonous”.

R2 51. you mean that it is challenging to raise cattle if ragwort is abundant?

A: We rephrased. 

R2 56. citation required

A: Citation added.

R2: 58. If the concern is that cattle will consume the biomass (either in the form of hay or as living plants), wouldn’t leaving litter in place also be risky to cattle?

A: This is true. Thus, we would recommend farmers to remove cattle from the pasture when ragwort is cut and for few weeks later. Thereafter, ragwort litter is decomposed and not poisonous anymore.

R2:64. here and throughout, be more specific. “changing” is too vague and does not illustrate the mechanism.

A: We added information on mechanisms.

R2:66. The mechanisms have not been sufficiently described. More detail is needed in how and why litter affects microclimate and nutrient availability, either within the context of ragwort specifically or herbaceous weeds in general.

A: We added information on mechanisms.

R2: 68. paragraph needs a topic sentence and clearer structure. We jump from addressing seedling establishment directly, to abiotic impacts of litter, to litter mixing effects. Break these concepts up into separate paragraphs and provide clear transition between each.

A: We reworked the whole paragraph. 

R2: 83. Is it intuitive? What about allee effectS?

A: Allee effects occur when populations reach a critical minimum size. As ragwort populations in our research area are very vivid and ragwort as a ruderal species is adapted to start new populations out of few seeds, we do not consider allee effects as an important for ragwort here. 

R2:84. assumed by whom?

A: Reference is given.

R2:88. Is it just this one study or were there more?

A: There were more studies. Now cited.

R2: 89. Incomplete sentence – what did they show about litter and soil?

A: Sentence rephrased.

R2: 90. What is the feedback? Is it intra-specific or inter-specific? Provide detail.

A: Intraspecific was added

R2: 90. “differences” is too vague

A: Mechanism was added 

R2: 91. is “probably driven by differences in the community” your speculation or that of the cited works?

A: speculation by cited work

R2: 92. Negative effects on which response?

A: Ragwort performance (biomass). Detail was added

R2: 95. the introduction should establish which factors are important to seedling establishment and seedling emergence as well as highlight differences between the two. Currently no support for this hypothesis is presented.

A: In line 74-84 we shortly discuss factors for seedling establishment and emergence. Furthermore, the support for this hypothesis is shown by the cited reference (Loydi et al. 2014).

R2: 103. In general, this section should not be necessary. It should be enough to state the hypotheses or questions without going back and explaining each. At the point that the hypotheses are presented, the reader should already be familiar enough with the mechanisms to understand the hypotheses without explanation. The current format of presenting a question and then detailing the predicted response is confusing and seems to be used instead of adequate background information beforehand. Furthermore, there are no citations presented within these explanations, so it is unclear what is hypothesis and what is already known.

A: We disagree; the comment of Reviewer 1 supports this by explicitly praising our clear hypothesis. The current format allows us to directly formulate our research hypothesis, which is derived from all the introduction written before the hypothesis are phrased. Thus, we have a clear section that sums up our assumptions based on all the information gathered in the introduction. Therefore, no further citations are included in this section. The current format allows us to build a frame for the study and is a clear guide to the reader, which questions he or she can expect to be answered.

R2:107. What do you mean by “congruent” – that there will be no difference between litter types?

A: We replaced congruent with similar. 

R2: 114. Confusing. Break into multiple sentences.

A: We followed this advice.

R2: 124. This information would be more useful in the Introduction since the paper focuses on ragwort specifically. Also, additional information on the ecological effects of ragwort would be helpful.

A: We included all information necessary in the introduction already. This section allows us to explicitly portrait the studied species. “Information on ecological effects” of ragwort is rather vague. We could describe ragworts role as important habitat for insects or its positive soil feedback for other plants, but this would not be an important information for the current study. Thus, we refrained from it.

R2: 144. Schleswig-Holstein, Germany

A: We followed this advice.

149. Provide GPS coordinates

A: We followed this advice.

R2: 151. The soil is still conditioned, just not by ragwort. It would be more accurate to describe it as "grass-conditioned" rather than "unconditioned", assuming that the dominant species where soil was collected were grasses.

A: It is true that “unconditioned” soil as we phrase it, is truly conditioned by other plants than ragwort (including grasses and other forbs). However, as ragwort and effects of its litter are in the focus of our study and not “grasses and other forbs”, we find it more convenient to refer to ragwort conditioned soil as “conditioned” and not ragwort conditioned soil as “unconditioned”. We clarified this definition in our method-section.

R2: 153. That is a relatively low-intensity drying treatment. Did you verify that litter had reached constant mass by that point?

A: To dry litter at low temperatures allowed us to preserve all possible influential chemicals within the litter, which are part of the complex of litter effects. Still, we had the same concern you stated in earlier litter experiments; as a consequence we tried to assess how much more additional weight litter loses when dried at higher temperatures (110°C ) some years ago; obviously the additional loss varied with the litter type but in most cases this loss was well below 5% and in no case above this value (not published personal observation). 

R2: 154. How do these litter treatments correspond to natural conditions?

A: This has been explained in the manuscript (line 158).

R2: 155. This is a weakness of the experimental design that the authors must address. The study focuses on specific litter effects, and while ragwort is maintained as a specific treatment, the grass litter treatment is poorly controlled. The exact composition of the grass litter treatment is unclear. Ideally, this would have been a single grass species, but instead appears to be a presumably variable mixture of different species. This severely limits the authors’ ability to detect litter mixing effects since the baseline additive condition would vary with the composition of the grass litter. This is true for all litter experiments to some extent since litter quality is variable even within a single plant, but by including multiple grass species as a single treatment, this error is enhanced.

A: In fact, using a litter mixture as comparison is a strength rather than weakness as it mirrors natural conditions we wanted to mimic. For our research question we wanted a comparison of litter without ragwort, litter of pure ragwort and a mix of the two. We were not interested in pure grass effects as this does not lead to an answer of the evaluation if ragwort litter can act as establishment barrier for ragwort. As we found clear effects of grass litter vs. ragwort litter despite using the rather uncontrolled grass treatment, actually supports the clearance and strength of the patterns we found.

R2: 158. Where are data from these pots presented, or how were they considered in the analyses? If litter type and amount affects germination, wouldn’t you also need to expose control pots to the same conditions to accurately account for seedbank under the considered conditions?

A: Control pots for accounting for seedlings from the seed bank where used to control for the probability of extra-emerging seeds from the seed bank. As no seedlings emerged no corrections of seedling numbers were needed.

R2:165. When was litter applied?

A: At setup of the experiment. Information was added.

R2: 214. This is a 56% increase, not a 33% increase. This is also somewhat confusing because your analyses were using LnRR and not on the actual observed data. So, when we look at Fig 1, we do not see a 56% increase, we only see that ragwort litter suppressed emergence and grass stimulated emergence.

A: We corrected this.

R2: 215. “interestingly” is not needed

A: We deleted this.

R2: 216. Unclear. What do you mean by “opposite?” Where is this shown? Was there a species by amount interaction – Fig 1 shows no effect of 2 g/dm2 of litter.

A: We rephrased this part.

R2: 217. There is an over-reliance on “i.e.” throughout the manuscript. Be concise. 

A: We corrected this.

R2:218. Also worth noting that 2 g/dm2 had no impact.

A: Changed according to new analysis.

R2: 219. MSQ is not necessary since you have provided F statistics

A: This information might seem superfluous on first sight. Still, it helps the reader to follow the statistical analyses, which is not the case when only the F-value is presented. Therefore, we would like to keep the MSQ in the table.

R2: 236. all interactions are quantitative – overly wordy. Erroneous bold.

A: Interactions can either be quantitative (seize of the effect differs) or qualitative (sign/direction of the effect changes). We want to stress that the direction (sign of the effect: low litter amounts leading to higher seedling mass) is the same we for all litter types. Thus, we want to keep this point. 

R2: 238. Fig 2 does not support this since grass litter error bars overlap and do not differ from 0 (controls).

A: This is true and already discussed in the very same sentence of the original manuscript. But the whole paragraph has been reworked. 

R2: 241. This appears to be redundant with line 239.

A: The whole paragraph has been reworked.

R2: 255. new paragraph

A: The whole paragraph has been reworked.

R2: 259. Quantify change. This is an issue throughout the results.

A: The whole results part has been reworked according to this critique.

R2: 289. But you also only tested 2 levels of litter. How well these levels correspond to real world conditons as well as the conditions tested by those other studies is an important consideration.

A: True, but this does not change the validity of the statement. How levels correspond to natural conditions was already described earlier (line 155-159).

R2: 290. You did not test for autotoxicity and cannot directly support this claim. There are other factors that could explain the presented trends.

A: We do not quite understand this claim. We tested for autotoxicity, since we compared germination and establishment of seeds of a species within its litter and control litter. Other mechanisms than autotoxicity are explicitly referred to in the discussion including a change in light, temperature, and water regime (line 331ff, 387ff, 429ff). 

R2: 296. Given that litter did not consistently cover the soil surface, does that mean that using mass per area is actually a fair method for comparing litter types? Type is confounded by cover (even if mass per area is constant).

A: Type will always be confounded by cover. Nevertheless, using mass is still a valid approach. But since we were aware of these difference in cover caused by the same amount of different litter, we discussed this (line 333ff).

R2 304. which grass litter? Be specific.

A: Alopecurus pratensis was added.

R2: 308. But ragwort had lower germination than controls, which had no litter at all. So, even though ragwort decayed faster than grass litter, it should still have a neutral-positive impact relative to controls.

A: This is true. Therefore, we assume that litter leachate from ragwort also negatively influences germination and added this point subsequently. 

R2: 322. Not exactly. Even if grasses and ragwort interacted, we would generally expect an intermediate response. What matters is if that response differs from the average of grasses and ragwort (assuming a 1:1 mixture grass:ragwort). The available data could easily test this.

A: We adjusted our analysis and added information on whether effects were additive or not. To this end, we used contrasts to specifically test whether the effect of mixed litter on response variables was mid-way between that of grass and ragwort alone (additive effect of pure type litters in the mixture).

R2: 326. That depends on the cover of ragwort (and on whether there was a mixing effect). If there is a perfect mixture of grasses and ragwort, your results suggest a neutral effect. If ragwort is abundant, then there would be a negative impact of leaving litter on ragwort germination. 

A: This is true. But even if germination is hampered, seedling establishment is not. Thus, the net effect is not clear. We stated this in our conclusions.

R2: Also, is ragwort litter commonly removed by burning or raking? 

A: No, as this is very laborious, ragwort litter is usually left on the grasslands.

R2: It is important to note that litter has different physical and chemical properties than green biomass, so this isn’t exactly comparable to land manager mowing and leaving the trimmings in field.

A: This depends on the management. As we work with an authority, that wants to reduce insect damage by mowing, a bar-mower instead of a mulcher is used. This machine is comparable to a scythe. Therefore, the resulting “real world” litter is comparable to our litter. Furthermore, we expect the basic patterns for green biomass and litter to be comparable and can be used as a first approach to find an answer to topic.

R2: 341. If the biomass result is primarily a consequence of competition, is it fair to ascribe it to the litter type in the first place?

A: Yes, since litter type does influence seedling density in the first place.

R2: 349. Yes, but it takes time for these nutrients to be converted into forms usable by plants

A: We added this point.

R2: 414. See comments above – this is not a robust conclusion.

A: We tried to put our conclusions in perspective to make them more robust.

R2: Fig 1. Increase axis label size. Why are groups using upper case letters here and lower case letters in Fig 2? Unify if not intentional.

A: Lower case letters were used when there is an interaction like in Fig. 2. Upper cased letters are used for main effects. We added this information in the Figure descriptions.

R2: Fig 2. Add legend to figure. Increase axis label size.

A: Explanations are given in the figure caption.

R2: Fig 3. Increase axis label size.

A: We increased labels.

R2: Figs 1,3,4 – It would be helpful to have all of these presented in the format of Fig 3 with type, amount, and soil presented in Fig 1 and 4 as well. Even though soil was not significant in figs 1 and 4, it would provide consistency across a somewhat complex set of analyses and would make comparisons easier for the reader.

A: Figures are presented in the same format when no interaction of litter type and amount occurred. Soil is always included in the control displayed in all figures.

---

## [Decision Letter · Decision Letter 1]

9 Dec 2020

PONE-D-20-12615R1

Conspecific and heterospecific litter effects on seedling emergence and growth in ragwort (Jacobaea vulgaris)

PLOS ONE

Dear Dr. Möhler,

Thank you for submitting your manuscript to PLOS ONE. After careful consideration, we feel that it has merit but does not fully meet PLOS ONE’s publication criteria as it currently stands. Therefore, we invite you to submit a revised version of the manuscript that addresses the points raised during the review process.

We look forward to receiving your revised manuscript.

Kind regards,

Craig Eliot Coleman, PhD

Academic Editor

PLOS ONE

Reviewers' comments:

Reviewer's Responses to Questions

**Comments to the Author**

1. If the authors have adequately addressed your comments raised in a previous round of review and you feel that this manuscript is now acceptable for publication, you may indicate that here to bypass the “Comments to the Author” section, enter your conflict of interest statement in the “Confidential to Editor” section, and submit your "Accept" recommendation.

Reviewer #1: All comments have been addressed

Reviewer #2: (No Response)

2. Is the manuscript technically sound, and do the data support the conclusions?

Reviewer #1: Yes

Reviewer #2: Yes

3. Has the statistical analysis been performed appropriately and rigorously? 

Reviewer #1: Yes

Reviewer #2: Yes

4. Have the authors made all data underlying the findings in their manuscript fully available?

Reviewer #1: Yes

Reviewer #2: Yes

5. Is the manuscript presented in an intelligible fashion and written in standard English?

Reviewer #1: Yes

Reviewer #2: Yes

6. Review Comments to the Author

Reviewer #1: The authors have undertaken a comprehensive revision to address my main points and those of the other Reviewer. I feel that my main points of critique have been addressed. The language is clearer, though I did make more suggested edits for clarity and efficiency in the attached manuscript. I have two main remaining suggestions to this revised manuscript.

First, I appreciate the expanded presentation of the Control results, but I could not sort out what the controls represented. Were they controls for litter addition, in which seeds were added without litter? The Methods state that “no ragwort seedlings emerged (Line 178)”, and yet the authors report emergence, biomass, etc. results for the Controls. Remind the reader in the Results what they are controls for – presumably for the effect of litter?

Second, I had trouble keeping the study’s purpose in mind as I read most of the Discussion. I appreciated your strong conclusion about the implications for management, but that question should be highlighted much earlier in the Discussion. That would provide the reader context for the details about emergence versus biomass that follows.

Further comments – mostly at the sentence-level, can be seen in the attached manuscript.

Dr. Jeffrey Corbin

Union College

Reviewer #2: Comments to the Authors

The manuscript is much improved and I appreciate the authors’ thoughtful responses to my earlier comments. In particular, the re-imagining of the statistical analyses makes the paper much more intuitive and also provides some new, exciting results. Although I still have some substantial concerns, I think they are fairly easily-addressed.

This study doesn’t strictly test the effects of leachates since the presence of leachates are confounded with the presence of other chemical and physical impacts of litter. This point is now emphasized throughout the discussion, and I think the authors do a good job explaining the importance of these other mechanism for their results. However, their argument and their data (see specific comments) suggest a subsidiary role of autotoxic litter leachates in ragwort performance, which is in contrast to the narrative established in the introduction. Therefore, a more consistent narrative could be created by emphasizing physical impacts of litter in the introduction.

A more direct test of the effects of leachates would be to isolate leachates and artificially impose them on soils/seeds in the absence of litter itself. Since this is not done and the authors make a strong argument for physical mechanisms, I remain confused by the assumptions made in the intro and the discussion that leachates are a primary cause of the observed results. The authors should review the manuscript to ensure it reflects their confidence in the various proposed mechanisms.

I question the accuracy of framing this study using the hetero- con-specific dichotomy. This is not a highly-controlled test of conspecific litter impacts relative to heterospecific litter since the non-ragwort litter was derived from very different species (grasses). This means that differences in the “conspecific” result and the “heterospecific” result could be due to any number of traits that differ between the species, not just species identity. For example, the design does not allow the authors to identify how much of the “conspecific” effect is due to seeds and litter being from the exact same species versus that litter belonging to any forb in general. It may be that other forbs are relatively rare and so “heterospecific” and “grass” may be functionally interchangeable in this study system, but that is not true everywhere. Especially given the broad scope of the journal, this should be addressed in the manuscript and/or mentions of “heterospecific” should be changed to “grass” (including the title).

I kept expecting a type of synthesis of the results from an invasion ecology or successional perspective that never quite landed. The results suggest a positive feedback loop to invasion (similar to findings of Schuster Dukes 2014 Oikos). Initially, sites have high biotic resistance against ragwort invasion since grass litter suppresses growth. However, since germination under grass litter is enhanced, they eventually establish. As ragwort start to accumulate on site, the non-additive interactions observed in growth take effect – facilitating enhanced growth of ragwort in the early and mid stages of invasion. This enhanced growth allows them expedite the competitive exclusion of grasses and dominate the site. Once ragwort becomes dominant and there is less grass litter on site, litter interactions start to wane but increasingly pure ragwort litter means bigger and bigger ragwort. Eventually grasses are excluded and ragwort is a self-sustaining monoculture. In short: differing impacts of grass and ragwort litter mean that ragwort benefits from whatever litter is there (grass promotes establishment early in invasion, ragwort-grass mixtures promote competitive ability mid-invasion, and ragwort promotes exclusion of other species late in invasion). The points are there throughout the discussion, but I think the authors could make it a concise point in the conclusion or near the end of the discussion. This feedback loop suggests that managers should limit ragwort litter on site.

A less critical point: Throughout the introduction, it is unclear which literature pertains to ragwort specifically. Occasionally, ragwort will be mentioned, and so these are presumably the only studies that consider ragwort, but it is somewhat jarring to continually switch focus from broad to specific throughout the section.

The manuscript requires minor editing for English.

Specific comments

50. Siberia is part of Asia

53. It is worth pointing out that these occur in both its native and invasive ranges. Is it necessary to list out the specific acts for all of these countries – can they not just be cited?

58. The remainder of this paragraph is a little hard to follow – look for ways to be more concise.

61. It is unclear that these allelopathic effects originate from litter at this point (I tend to think first of root exudates), making this hard to follow at first.

64. This is a hard sell at this point since the first half of the paragraph talks about why leaving biomass in field is typically not done (toxicity to livestock and seed mass). It may be more effective to bring these drawbacks up later instead of leading with them.

65. “important” “prominent” are vague and somewhat redundant with following sentences.

67. this is explained later – drop

71. this is more of a continuous effect than an threshold effect as is implied by “too high.” Incomplete sentence.

75. Unnecessary to call this out specifically – most seeds are photo-period and/or moisture sensitive – work reference into following sentences.

82. These are much less intuitive mechanisms, and yet they receive only a fraction of the attention. I don’t think you need to go into more detail on these, but the discussion on lines 75-81 could be condensed to be of similar brevity.

86. by the presence of litter

87. what is “it”

90. But forb litter also often decomposes more quickly thereby reducing physical mechanisms of seedling suppression

97. Not necessarily

98. unclear. “was found” is unnecessary – reword to something like “can be.”

102. redundant with 101.

109. overly vague and confusing. State simply that germination may be non-additively reduced by concurrent litter cover and soil conditioning.

116. litter amount

117. to what end? Reword.

118. Heterospecific litter has not been mentioned by name until this point. Litter effects in general have been discussed, and the added impacts of conspecific litter have been discussed, but you have not made the point that heterospecific litter may act vastly different.

121. Delete “assuming conspecific negative effects” – redundant with what follows

123. Earlier you contrasted grass and forb litter – so to what extent are the observed impacts of ragwort due to it being conspecific litter versus it just being forb litter? Would a different forb have comparable effects? What about a native congeneric?

124. But don’t you say earlier that they could grow more due to lower competition (opposite of what is said here)?

130. “We expect the interaction to be lowest” – unclear

131. Is this correct? “higher” under medium ragwort and “highest” under low grass – what has the lowest performance? By “ragwort” you mean “pure ragwort” and by “grass” you mean “pure grass,” correct?

136. delete “as negative… observed.”

166. I still think it is somewhat misleading to call this “unconditioned” since it isn’t soil that was sterilized other otherwise treated (and is therefore conditioned, just not by ragwort). My concern is alleviated by the emphasis placed here on where and how the soils were treated, but maintaining that these soils are “unconditioned” puts the burden on the reader to keep this caveat in mind.

174. dm is not SI

179. This is true of all the pots, not just the “control” pots?

186. What is the approximate mesh size of the grid?

190. tallest plant

227+ break this into two paragraphs based on factor (do same for growth)

234. stats show this is significant, delete “significantly’

235. So about 25% greater than ragwort?

238. This is a different unit than used in methods. Use gm2 throughout.

259. that is interesting.

284. also interesting

308. A microclimate effect (higher moisture allows for greater stomatal conductance via more leaves)?

310. these results based on the controls are valuable since they highlight how variable the controls were – I think this points to the current analyses being well-structured and insightful.

321. This shouldn’t be just “interesting to note,” it should be (or is) one of the most important takehome messages.

324. variance in which response?

326. species-specific

328. Yes, but there are many other possible explanations too. Litter impacts on microclimate, the rate of litter decomposition and N release, and the relative difference between different litter amounts compared to differences between litter types could all be drivers of this result. There is no way to exclude these other factors given the current design, so there is little reason to expect that autotoxicity alone is the primary factor.

330. “no litter” is control, right? Worth clarifying.

332. “ high and/or ragwort” is awkward. Maybe “high amounts of litter and/or ragwort litter”

351. This was a good explanation of the physical impacts of litter on soil conditions and how those differ based on the decomposition rates of grasses and forb litter. However, those physical impacts are not leachate effects, so it is confusing to say “leachate effects may also depend…”

356. IF leachate effects were present. I think the soil moisture impacts are at least equally as likely given your arguments above and below. Do you mean “speculate” instead of “assume”?

370. Does the lack of a conditioning effect suggest a lack of a chemical/autotoxic effect of litter as well? Presumably leachates would accumulate in soils in such a way that you would detect a legacy effect of leachates in your conditioning analyses. In contrast, there would not be the same legacy effect of a physical mechanism, therefore fitting with the lack of a conditioning effect.

388. delete “rather productive”

389. This makes sense: when there is not a monoculture of ragwort, litter facilitates invasion. When there is a monoculture, litter slightly inhibits invasion. So, remove litter when there are grasses still present, but leave litter alone when ragwort is the only thing left. The question then though is if this even matters – once there is a ragwort monoculture, it doesn’t matter if germination is reduced so long as the population can maintain itself. The reduction in germination rates would have to be large enough to make the population collapse or for the establishment of later-successional species. I think this effect is outweighed by the benefits of larger ragwort under ragwort litter - this allows them to exclude other species from the site (so even with lower germination they still win).

389. “when litter cover is to high,” and afterward is unclear.

403. Right, but see above (does this matter for land managers?)

420. important point

450. I don’t think there is strong evidence of autotoxicity based on this study alone. In contrast, there are several lines of data/observations pointing at a physical mechanism. So, “most likely” might not be the most accurate.

462. Congratulations on a strong revision.

Figures: increase axis label size. Position letters above whiskers.

+

7. PLOS authors have the option to publish the peer review history of their article (what does this mean?). If published, this will include your full peer review and any attached files.

Reviewer #1: **Yes: **Jeffrey D Corbin

Reviewer #2: No

---

## [Author Response · Author response to Decision Letter 1]

18 Jan 2021

Dear Editor and Reviewers,

We thank you very much for your time, efforts and helpful comments and critiques. Based on both reviews we created an updated and improved manuscript.

Further down, we would like to directly address these points, with your critique stated repeated (following R1/R2) and our answer (following A). Minor points were track changed directly in the manuscript.

In any case, we thank you for taking our manuscript into consideration anew and thank you again for your helpful critique and comments.

Yours sincerely

Henrike Möhler

Reviewer 1:

R1: I have two main remaining suggestions to this revised manuscript.

First, I appreciate the expanded presentation of the Control results, but I could not sort out what the controls represented. Were they controls for litter addition, in which seeds were added without litter? The Methods state that “no ragwort seedlings emerged (Line 178)”, and yet the authors report emergence, biomass, etc. results for the Controls. Remind the reader in the Results what they are controls for – presumably for the effect of litter?

A: We added information on the control pots. Indeed, the pots with no seedlings were “extra-control pots” that did not receive any seeds.

Second, I had trouble keeping the study’s purpose in mind as I read most of the Discussion. I appreciated your strong conclusion about the implications for management, but that question should be highlighted much earlier in the Discussion. That would provide the reader context for the details about emergence versus biomass that follows.

A: In the start of the discussion we highlighted the management advice we would give based on our study

Reviewer 2:

R2: This study doesn’t strictly test the effects of leachates since the presence of leachates are confounded with the presence of other chemical and physical impacts of litter. This point is now emphasized throughout the discussion, and I think the authors do a good job explaining the importance of these other mechanism for their results. However, their argument and their data (see specific comments) suggest a subsidiary role of autotoxic litter leachates in ragwort performance, which is in contrast to the narrative established in the introduction. Therefore, a more consistent narrative could be created by emphasizing physical impacts of litter in the introduction.

A: It is true that we did not test effects of pure leachate as these effects are confounded with physical litter aspects. However, we did a pure leachate test earlier and found autotoxic effects. As far as we understand you question effects of leachate in general. However, we see that seedlings emerge less often under ragwort litter and we know from earlier experiments from us and others (Van de Voorde et al) that there is an autotoxic effect. Hence leaving out autotoxicity as an explanation would not give a full picture. Therefore, we restrain from changing the narrative in the introduction, as we feel that both parts: allelopathy and physical effects should be given a fair share in the narrative of the introduction and discussion.

We checked the whole text again to find passages where we overemphasized the importance of leachate and changed the wording in those parts. However, we do not feel that the narrative in general is overemphasizing chemical effects. 

R2: A more direct test of the effects of leachates would be to isolate leachates and artificially impose them on soils/seeds in the absence of litter itself. Since this is not done and the authors make a strong argument for physical mechanisms, I remain confused by the assumptions made in the intro and the discussion that leachates are a primary cause of the observed results. The authors should review the manuscript to ensure it reflects their confidence in the various proposed mechanisms.

A: As you do, we do not assume chemical reasons as primary cause of our results. Maybe we overemphasized the importance of the litter leachates. We rephrased respective parts accordingly.

R2: I question the accuracy of framing this study using the hetero- con-specific dichotomy. This is not a highly-controlled test of conspecific litter impacts relative to heterospecific litter since the non-ragwort litter was derived from very different species (grasses). This means that differences in the “conspecific” result and the “heterospecific” result could be due to any number of traits that differ between the species, not just species identity. For example, the design does not allow the authors to identify how much of the “conspecific” effect is due to seeds and litter being from the exact same species versus that litter belonging to any forb in general. It may be that other forbs are relatively rare and so “heterospecific” and “grass” may be functionally interchangeable in this study system, but that is not true everywhere. Especially given the broad scope of the journal, this should be addressed in the manuscript and/or mentions of “heterospecific” should be changed to “grass” (including the title).

A: We used the hetero- and conspecific dichotomy since the effects of ragwort on itself (conspecific) were in the focus of our study and using this word gives the reader the concrete origin of the litter as rather con- or heterospecific. Your right, that other forbs may have similar effects as ragwort. Therefore, we added “grass” to heterospecific. We hope that this is a compromise were the reader can see at one view that conspecific effects of ragwort are in the focus of our study compared to effects of other litter, which consisted mainly of grass.

R2: I kept expecting a type of synthesis of the results from an invasion ecology or successional perspective that never quite landed. The results suggest a positive feedback loop to invasion (similar to findings of Schuster Dukes 2014 Oikos). Initially, sites have high biotic resistance against ragwort invasion since grass litter suppresses growth. However, since germination under grass litter is enhanced, they eventually establish. As ragwort start to accumulate on site, the non-additive interactions observed in growth take effect – facilitating enhanced growth of ragwort in the early and mid stages of invasion. This enhanced growth allows them expedite the competitive exclusion of grasses and dominate the site. Once ragwort becomes dominant and there is less grass litter on site, litter interactions start to wane but increasingly pure ragwort litter means bigger and bigger ragwort. Eventually grasses are excluded and ragwort is a self-sustaining monoculture. In short: differing impacts of grass and ragwort litter mean that ragwort benefits from whatever litter is there (grass promotes establishment early in invasion, ragwort-grass mixtures promote competitive ability mid-invasion, and ragwort promotes exclusion of other species late in invasion). The points are there throughout the discussion, but I think the authors could make it a concise point in the conclusion or near the end of the discussion. This feedback loop suggests that managers should limit ragwort litter on site.

A: We feel uneasy to resume the proposed feedback loop. From a successional point of view, ragwort is an early successional that profits from open soils (as mentioned in the methods section). It never establishes in grassland with intact sward and high litter but needs disturbed ground for germination. If open soil patches are available and a light litter cover protects the seeds and seedlings ragwort will establish. But it will never form monocultures without grasses and other forbs. When the soil is not disturbed grasses will outcompete ragwort sooner or later at least in our study system where ragwort is native. The system we studied is much more complex in the field than the feedback loop you proposed, especially in managed grasslands. In our simple experiment we did not find univocal effects of ragwort litter on its germination and establishment. Germination was higher under ragwort, but resulting seedlings were stronger. If even our simple experiment does not show a clear negative effect of ragwort litter on both germination and growth, it is rather unlikely to find distinct effects in the field. Thus, we refrain from proposing your synthesis as we do not see this mirrored in the natural system.

R2: A less critical point: Throughout the introduction, it is unclear which literature pertains to ragwort specifically. Occasionally, ragwort will be mentioned, and so these are presumably the only studies that consider ragwort, but it is somewhat jarring to continually switch focus from broad to specific throughout the section.

A: We tried to find as much literature that dealt specifically with ragwort as possible. However, some aspects are only studied on other plants or assumed to be of general importance for all forbs. Thus, we feel that combining both general and ragwort-specific literature will give a more consistent picture than focusing only on “general” or “ragwort” studies. Where it is important that we directly refer to ragwort this is mentioned in the text. In other parts we did not clarify that as it made no difference in the point, we wanted to make but would rather complicate the text for the reader. 

R2: The manuscript requires minor editing for English.

Specific comments

R2: 50. Siberia is part of Asia

A: True. Corrected.

R2: 53. It is worth pointing out that these occur in both its native and invasive ranges. Is it necessary to list out the specific acts for all of these countries – can they not just be cited?

A: Laws were omitted.

R2: 58. The remainder of this paragraph is a little hard to follow – look for ways to be more concise.

A: We reformulated parts of the paragraph to be more concise.

R2: 61. It is unclear that these allelopathic effects originate from litter at this point (I tend to think first of root exudates), making this hard to follow at first.

A: We stated more precisely “litter leachates”.

R2: 64. This is a hard sell at this point since the first half of the paragraph talks about why leaving biomass in field is typically not done (toxicity to livestock and seed mass). It may be more effective to bring these drawbacks up later instead of leading with them.

A: Even though it might not be wise to leave biomass on the field for the reasons mentioned, it is nevertheless typical to leave the biomass as it is very time consuming and costly to remove the litter. We added this point in the introduction.

R2: 65. “important” “prominent” are vague and somewhat redundant with following sentences.

A: We changed the wording accordingly.

R2:67. this is explained later – drop

A: True. Corrected.

R2: 71. this is more of a continuous effect than an threshold effect as is implied by “too high.” Incomplete sentence.

A: We reformulated.

R2: 75. Unnecessary to call this out specifically – most seeds are photo-period and/or moisture sensitive – work reference into following sentences.

A: True. Corrected.

R2: 82. These are much less intuitive mechanisms, and yet they receive only a fraction of the attention. I don’t think you need to go into more detail on these, but the discussion on lines 75-81 could be condensed to be of similar brevity.

A: True. Corrected.

R2: 86. by the presence of litter

A: We didn’t understand this remark

R2: 87. what is “it”

A: “Litter type” replaces “it”.

R2: 90. But forb litter also often decomposes more quickly thereby reducing physical mechanisms of seedling suppression

A: True. We added this remark.

R2: 97. Not necessarily

A: We reformulated.

R2: 98. unclear. “was found” is unnecessary – reword to something like “can be.”

A: We reformulated.

R2: 102. redundant with 101.

A: We omitted that part.

R2: 109. overly vague and confusing. State simply that germination may be non-additively reduced by concurrent litter cover and soil conditioning.

A: We omitted that part.

R2: 116. litter amount

A: We changed the order of words if this was your critique.

R2: 117. to what end? Reword.

A: reworded.

R2: 118. Heterospecific litter has not been mentioned by name until this point. Litter effects in general have been discussed, and the added impacts of conspecific litter have been discussed, but you have not made the point that heterospecific litter may act vastly different.

A: True. We added this point in the introduction.

R2: 121. Delete “assuming conspecific negative effects” – redundant with what follows

A: deleted.

R2: 123. Earlier you contrasted grass and forb litter – so to what extent are the observed impacts of ragwort due to it being conspecific litter versus it just being forb litter? Would a different forb have comparable effects? What about a native congeneric?

A: As former studies (pure leachate studies) showed that ragwort is autotoxic, we assume that conspecific effects differ from heterospecific effects. It is certainly true that other especially congeneric species may also hamper seedling emergence in ragwort as they contain similar chemical compounds. And it is an interesting question what effect native congeneric species would have. However, the scope of our study was to test whether ragwort litter can be used as a management tool and not to test if other congeneric forbs may also hamper its growth. 

R2: 124. But don’t you say earlier that they could grow more due to lower competition (opposite of what is said here)?

A: True. Corrected.

R2: 130. “We expect the interaction to be lowest” – unclear

A: Omitted.

R2: 131. Is this correct? “higher” under medium ragwort and “highest” under low grass – what has the lowest performance? By “ragwort” you mean “pure ragwort” and by “grass” you mean “pure grass,” correct?

A: Not necessarily. We want to express that seedling performance is worst in (pure) ragwort and high amounts of litter, regardless if pure or mixed. Further, we expected performance to be better in medium litter amounts regardless of pure or mixed ragwort and to be best in low litter amounts of pure grass litter. Therefore, we added “pure” for gras.

R2: 136. delete “as negative… observed.”

A: deleted.

R2:166. I still think it is somewhat misleading to call this “unconditioned” since it isn’t soil that was sterilized other otherwise treated (and is therefore conditioned, just not by ragwort). My concern is alleviated by the emphasis placed here on where and how the soils were treated, but maintaining that these soils are “unconditioned” puts the burden on the reader to keep this caveat in mind.

A: We explain in detail what we defined as “unconditioned”. Rephrasing “unconditioned” in grass-conditioned would miss the point as there are also other forbs that may conditioned the soil. Another wording such as ragwort and non-ragwort conditioned would be more confusing for the reader as “ragwort-conditioned” is the focus. In our study it does not matter how the other soil is conditioned. Thus, we decided to keep the dichotomy as we established it.

R2 174. dm is not SI

A: m is SI. Prefixes such as “c” “d” or “k” are standardized, too.

R2: 179. This is true of all the pots, not just the “control” pots?

A: True. Corrected.

R2: 186. What is the approximate mesh size of the grid?

A: We added that information in the text.

R2: 190. tallest plant

A: True. Corrected.

R2: 227+ break this into two paragraphs based on factor (do same for growth) 

A: We changed that accordingly.

R2: 234. stats show this is significant, delete “significantly’

A: True. Corrected.

R2: 235. So about 25% greater than ragwort?

A: That’s right. We added that information.

R2:238. This is a different unit than used in methods. Use gm2 throughout.

A: True. Corrected.

R2: 259. that is interesting.

A: We also think so.

R2: 284. also interesting

A: We think so, too. 

R2: 308. A microclimate effect (higher moisture allows for greater stomatal conductance via more leaves)?

A: This is possible.

R2: 310. these results based on the controls are valuable since they highlight how variable the controls were – I think this points to the current analyses being well-structured and insightful.

A: Thanks for your impulse on that.

R2: 321. This shouldn’t be just “interesting to note,” it should be (or is) one of the most important takehome messages.

A: True. Corrected.

R2: 324. variance in which response?

A: We added that information.

R2: 326. species-specific

A: True. Corrected.

R2: 328. Yes, but there are many other possible explanations too. Litter impacts on microclimate, the rate of litter decomposition and N release, and the relative difference between different litter amounts compared to differences between litter types could all be drivers of this result. There is no way to exclude these other factors given the current design, so there is little reason to expect that autotoxicity alone is the primary factor.

A: True. We gave those explanations more space by mentioning same directly.

R2: 330. “no litter” is control, right? Worth clarifying.

A: True. Corrected.

R2: 332. “ high and/or ragwort” is awkward. Maybe “high amounts of litter and/or ragwort litter”

A: True. Corrected.

R2: 351. This was a good explanation of the physical impacts of litter on soil conditions and how those differ based on the decomposition rates of grasses and forb litter. However, those physical impacts are not leachate effects, so it is confusing to say “leachate effects may also depend…”

A: True. We omitted “leachate to clarify that.

R2: 356. IF leachate effects were present. I think the soil moisture impacts are at least equally as likely given your arguments above and below. Do you mean “speculate” instead of “assume”?

A: We discussed moisture effects earlier and later. Thus, they are given their fair share. Autotoxicity is another possible explanation that also should have its room here. We already emphasize that leachate effects are probably only present for a very short time. That’s what we assume.

R2: 370. Does the lack of a conditioning effect suggest a lack of a chemical/autotoxic effect of litter as well? Presumably leachates would accumulate in soils in such a way that you would detect a legacy effect of leachates in your conditioning analyses. In contrast, there would not be the same legacy effect of a physical mechanism, therefore fitting with the lack of a conditioning effect.

A: No. This was indicated with the notion that leachate effects are very short lived and do not translate into soil effects. We found leachate effects in a germination experiment and see the same patterns in the common-garden experiment: less seedlings emerge in ragwort litter. The chemicals in the leachates that cause this effect might be decomposed very fast in the soil or connect with other chemicals in the soil so that the lack of soil conditioning does not mean a lack of autotoxic effects of the leachate.

R2: 388. delete “rather productive”

A: deleted.

R2 389. This makes sense: when there is not a monoculture of ragwort, litter facilitates invasion. When there is a monoculture, litter slightly inhibits invasion. So, remove litter when there are grasses still present, but leave litter alone when ragwort is the only thing left. The question then though is if this even matters – once there is a ragwort monoculture, it doesn’t matter if germination is reduced so long as the population can maintain itself. The reduction in germination rates would have to be large enough to make the population collapse or for the establishment of later-successional species. I think this effect is outweighed by the benefits of larger ragwort under ragwort litter - this allows them to exclude other species from the site (so even with lower germination they still win).

A: As ragwort does not form monocultures this might be logical in theory but does not match the observation in the field. Grass is always present in “ragwort meadows”. Thus, we cannot adopt these assumptions. See also to the answer we were giving concerning the similar question in the general critique section.

R2: 389. “when litter cover is to high,” and afterward is unclear.

A: We detailed that part.

R2: 403. Right, but see above (does this matter for land managers?)

A: Yes, since already medium amounts of ragwort are problematic for farmers. 

R2: 420. important point

A: We did not understand what you wanted us to change here. 

R2: 450. I don’t think there is strong evidence of autotoxicity based on this study alone. In contrast, there are several lines of data/observations pointing at a physical mechanism. So, “most likely” might not be the most accurate.

A: We still think that autotoxicity is one explanation for less seedlings. However, we over-emphasized its importance here. We changed the wording accordingly.

R2: 462. Congratulations on a strong revision.

A: Thanks for your critical review! It improved the manuscript a lot.

R2:Figures: increase axis label size. Position letters above whiskers.

A: axis labels increased. Letters above whiskers.

---

## [Editor Report · Decision Letter 2]

20 Jan 2021

Conspecific and heterospecific grass litter effects on seedling emergence and growth in ragwort (Jacobaea vulgaris)

PONE-D-20-12615R2

Dear Dr. Möhler,

We’re pleased to inform you that your manuscript has been judged scientifically suitable for publication and will be formally accepted for publication once it meets all outstanding technical requirements.

Kind regards,

Craig Eliot Coleman, PhD

Academic Editor

PLOS ONE
---

## [Editor Report · Acceptance letter]

22 Jan 2021

PONE-D-20-12615R2 

Conspecific and heterospecific grass litter effects on seedling emergence and growth in ragwort (*Jacobaea vulgaris*) 

Dear Dr. Möhler:

I'm pleased to inform you that your manuscript has been deemed suitable for publication in PLOS ONE. Congratulations! Your manuscript is now with our production department. 

Kind regards, 

on behalf of

Dr. Craig Eliot Coleman 

Academic Editor

PLOS ONE